# Reversal of high-glucose–induced transcriptional and epigenetic memories through NRF2 pathway activation

Martí Wilson-Verdugo[1], Brandon Bustos-García[1], Olga Adame-Guerrero[1], Jaqueline Hersch-González[1], Nallely Cano-Domínguez[1], Maribel Soto-Nava[2], Carlos A Acosta[3], Teresa Tusie-Luna[4], Santiago Avila-Rios[2], Lilia G Noriega[5], Victor J Valdes[1]

**Diabetes complications such as nephropathy, retinopathy, or cardiovascular disease arise from vascular dysfunction. In this context, it has been observed that past hyperglycemic events can induce long-lasting alterations, a phenomenon termed "metabolic memory." In this study, we evaluated the genome-wide gene expression and chromatin accessibility alterations caused by transient high-glucose exposure in human endothelial cells (ECs) in vitro. We found that cells exposed to high glucose exhibited substantial gene expression changes in pathways known to be impaired in diabetes, many of which persist after glucose normalization. Chromatin accessibility analysis also revealed that transient hyperglycemia induces persistent alterations, mainly in non-promoter regions identified as enhancers with neighboring genes showing lasting alterations. Notably, activation of the NRF2 pathway through NRF2 overexpression or supplementation with the plant-derived compound sulforaphane, effectively reverses the glucose-induced transcriptional and chromatin accessibility memories in ECs. These findings underscore the enduring impact of transient hyperglycemia on ECs' transcriptomic and chromatin accessibility profiles, emphasizing the potential utility of pharmacological NRF2 pathway activation in mitigating and reversing the high-glucose–induced transcriptional and epigenetic alterations.**

## Introduction

Several diabetes-related pathologies, including nephropathy, retinopathy, and cardiovascular disease, are associated with vasculature alterations (Beckman & Creager, 2016). The vascular system is particularly sensitive to blood glucose concentrations, as endothelial cells (ECs) are in direct contact with the bloodstream and display increased susceptibility because of the predominant expression of insulin-independent glucose transporter GLUT-1 (Takata et al, 1997).

Hyperglycemia elicits a collection of molecular alterations in ECs; for instance, glucose-induced activation of NADPH oxidase via PKC induces the production of superoxide radicals and an increase in oxidative stress (Roberts & Porter, 2013). This is further exacerbated by the decreased activity of the antioxidant systems commonly observed in diabetes (Kowluru et al, 2007). In addition, the generation of advanced glycation end products (AGEs) promotes a pro-fibrotic state and activation of the RAGE receptor, which activates MAPK and NF-$\kappa$B pathways (Neumann et al, 1999; Ramasamy et al, 2008). This leads to the release of pro-inflammatory cytokines, such as IL-6 and TNF-$\alpha$, and the expression of adhesion molecules such as ICAM-1 and VCAM-1, ultimately leading to inflammation and vascular damage. Hyperglycemia-induced ROS also up-regulate RAGE (Yao & Brownlee, 2010), and impact the production of nitric oxide (Kolluru et al, 2012), leading to vasodilation alterations. Overall, these cumulative effects result in endothelial dysfunction, catalyzing the onset of diabetes-associated pathologies in the vasculature.

Longitudinal studies in diabetes patients revealed that past periods of hyperglycemia can have long-term detrimental effects (Nathan et al, 2005; Holman et al, 2008), a phenomenon currently known as "metabolic memory." This concept encompasses the observation that early and rigorous glycemic control leads to prolonged protection against diabetes complications, even years after discontinuing the intensive glycemic control (Lachin et al, 2021). A proposed mechanism underlying the metabolic memory posits a positive feedback loop where hyperglycemia induces overproduction of reactive oxygen species (ROS), which then induce and activate a myriad of pathways that converge in further ROS production (Ihnat et al, 2007). This causes continuous activation of the PKC and NF-$\kappa$B pathways (Giorgi et al, 2010; Lingappan, 2018), perpetuating a cycle of ROS-induced cell damage, dysregulation of gene expression, and prolonged inflammation even in the absence of hyperglycemia.

[1]Departamento de Biología Celular y del Desarrollo, Instituto de Fisiología Celular, Universidad Nacional Autónoma de México (UNAM), Ciudad de México, México   [2]Centre for Research in Infectious Diseases of the National Institute of Respiratory Diseases (CIENI/INER), Mexico City, Mexico   [3]Hospital Diomed, Ciudad de México, Mexico   [4]Unidad de Biología Molecular y Medicina Genómica Instituto de Investigaciones Biomédicas UNAM/Instituto Nacional de Ciencias Médicas y Nutrición Salvador Zubiran, Ciudad de México, Mexico   [5]Departamento de Fisiología de la Nutrición, Instituto Nacional de Ciencias Médicas y Nutrición Salvador Zubirán, Ciudad de México, Mexico

Correspondence: julian.valdes@ifc.unam.mx

Different studies demonstrated that metabolic memory can be replicated in vitro. Transient hyperglycemia in human-cultured cells induces a sustained increase in ROS, the overexpression of pro-inflammatory genes, and a diminished antioxidant response (Zhao et al, 2021; Yao et al, 2022). Epigenetic alterations are recognized as potential candidates in the establishment and persistence of glucose-induced cellular memory (Siebel et al, 2010; Okabe et al, 2012; Reddy et al, 2015). For example, transient hyperglycemia in aortic ECs triggers acquisition of active histone marks, while reducing repressive modifications on NOX4 and eNOS promoters, leading to sustained ROS production and vascular dysfunction (Liao et al, 2018). Furthermore, NF-κB overexpression is sustained by an antagonistic interplay between histone methyltransferases and demethylases after normalization of glucose, contributing to persistent inflammation (El-Osta et al, 2008; Brasacchio et al, 2009). Similarly, in renal ECs, a metabolic memory effect has been linked to renal dysfunction both in vivo and in vitro, linked to reduced chromatin accessibility, alterations in transcription factor binding, and diminished histone acetylation (Bansal et al, 2020). These alterations also correlate with increased DNA methylation at key genes involved in kidney transport. Moreover, in retinal ECs, hyperglycemia and ROS production can induce the sustained overexpression of DNA (cytosine-5)-methyltransferase 1 (DNMT1), even after normalization of glucose levels (Mishra & Kowluru, 2016), maintaining a pathological epigenetic memory. In addition, differences in DNA methylation and histone modifications have been observed in pro-inflammatory genes in blood cells of diabetic individuals where a metabolic memory has been established (Miao et al, 2014; Chen et al, 2016). Nonetheless, the study of the molecular mechanisms driving the establishment and persistence of the metabolic memory is still an active field of research.

Previous studies assessed potential interventions to prevent or "erase" the metabolic memory. For instance, Zhang et al showed that metformin or resveratrol supplementation in venous ECs prevented the glucose-induced increase in cellular senescence and that this effect was dependent on increased SIRT1 deacetylation of p53 (Zhang et al, 2015). More recently, Yao et al demonstrated that tBHQ-mediated activation of the nuclear factor erythroid 2 (NRF2)–related factor 2, a transcription factor that acts as the master regulator of the antioxidant and xenobiotic response in mammals, was able to revert transient high-glucose–induced sustained activation of the TGF-β and NF-κB pathways (Yao et al, 2022). In the same study, the authors found that treatment with tBHQ reverted collagen accumulation and perivascular fibrosis in a mouse model of metabolic memory. However, the potential beneficial effects of NRF2 activation at a transcriptome-wide scale are yet to be evaluated in the context of the glucose-induced metabolic memory.

Overall, evidence shows that transient periods of pathological hyperglycemia can imprint a "memory" or "legacy" effect in ECs. This memory encompasses increased inflammation, a surge in oxidative stress, and dysregulation of the genetic programs that ultimately compromise tissue homeostasis. This has important implications in diabetes-related vascular complications and highlights the need for new therapeutic strategies to counteract hyperglycemia's legacy effects.

In this study, we characterized the persistent genome-wide transcriptional and chromatin accessibility alterations induced by transient hyperglycemia in human ECs. Our findings reveal that most of the differentially accessible chromatin regions that occur after transient high glucose resulted in a gain of accessibility, with many of these overlapping with putative enhancers that presumably influence the expression of diabetes-related neighboring genes. Furthermore, our transcription factor motif analysis identified members of the basic leucine zipper (bZIP) family as potential players in establishing the glucose-induced transcriptional legacy response. In accordance, we demonstrated that the pharmacological activation of the NRF2 pathway reverted the glucose-induced transcriptional and chromatin accessibility memories in ECs. Our work contributes to a better understanding of the etiology of diabetes-associated vascular complications and aids in the development of strategies to mitigate the vascular damage associated with the metabolic memory.

## Results

### Transient high-glucose exposure induces a transcriptional memory in ECs

To explore the persistent effects of transient high-glucose concentrations in ECs, we conducted a series of experiments in HUVECs under different glucose concentrations. Our treatments comprised the culture of cells in three different treatments: normal glucose for 8 d (5.5 mM glucose; control), high glucose for 8 d (HG), and high glucose for 4 d followed by 4 d in normal glucose as our memory treatment (Fig 1A). To determine the glucose concentration to use for our HG treatment, we evaluated the changes in gene expression after exposing HUVECs to 15 or 30 mM glucose, as these concentrations can be found in uncontrolled patients with diabetes (Glaser et al, 1988; Ilkova et al, 1997). The genes evaluated belong to pathways known to be affected in diabetes, such as the TGF-β pathway, inflammation, and antioxidant response. We found that 4 d of 30 mM glucose induced substantial changes in gene expression compared with the control in four of the six genes evaluated, whereas 15 mM glucose only had effects in one gene after 8 d of treatment. Then, we assessed the endurance of a transient high-glucose episode by culturing cells for 4 or 8 d in normal glucose after exposing them to 30 mM glucose for 4 d (Fig 1B; "memory 4 d + 4 d" and "memory 4 d + 8 d," respectively). Our results showed persistent alterations in the expression of all six genes evaluated in the case of the memory 4 d + 4 d treatment, which persisted in three genes in the memory 4 d + 8 d treatment. These results prompted us to use 30 mM glucose for our HG treatment, and the memory 4 d + 4 d (henceforth referred to as "memory") treatment for the rest of our experiments involving the establishment and study of a "glycemic memory."

Then, as previous studies have reported alterations in cellular respiration and glycolysis in ECs in response to high glucose (Bertelli et al, 2022; Yao et al, 2022), we evaluated the glycolytic and respiratory rates in HUVECs exposed to our treatments including an

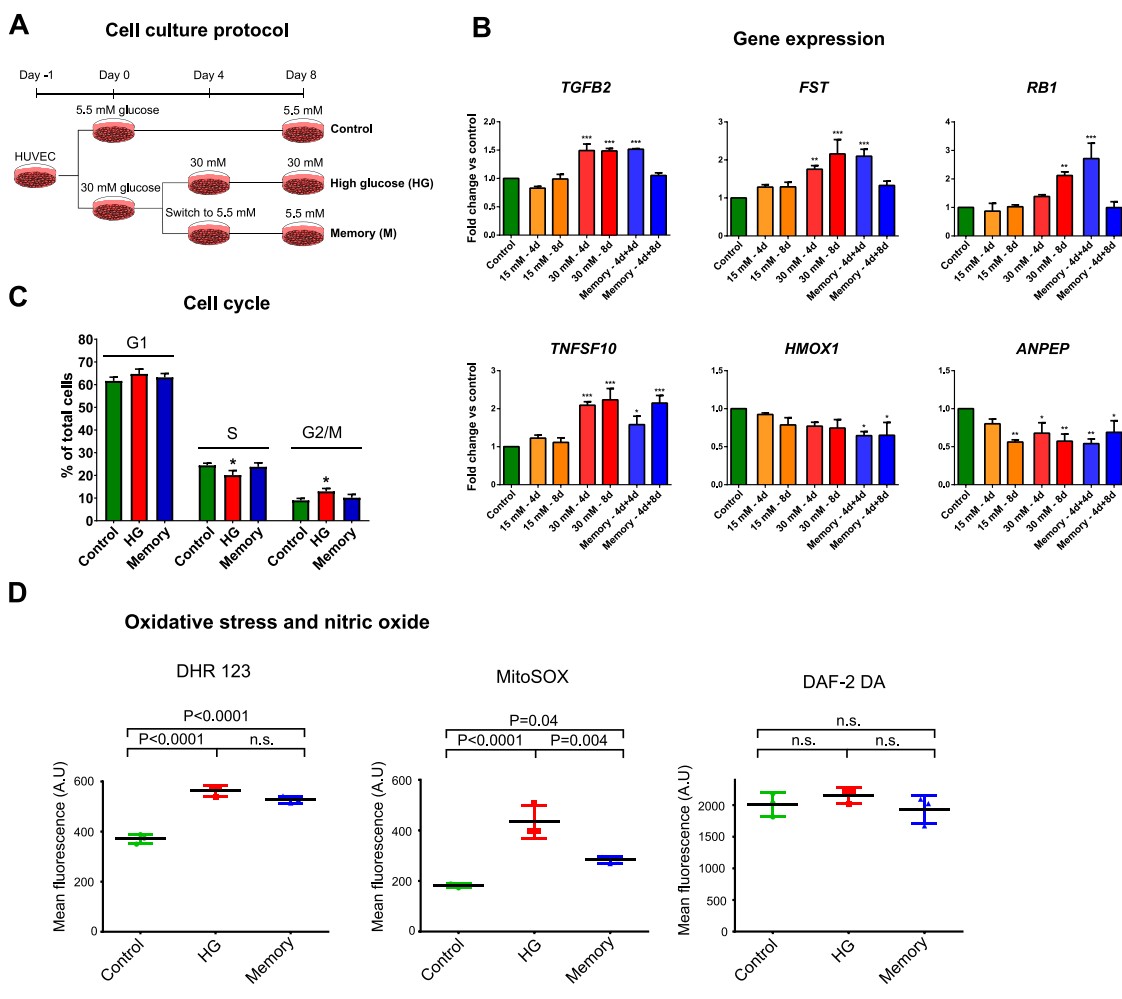

**Figure 1. High-glucose–induced transcriptional and oxidative memory in endothelial cells.**
**(A)** Experimental treatments in HUVECs: control (5.5 mM glucose, 8 d), high glucose (HG; 30 mM glucose, 8 d), and memory (4 d at 30 mM glucose, then 4 d in 5.5 mM glucose). **(B)** RT–qPCR of six genes in HUVECs exposed to different glucose treatments: control (5.5 mM glucose for 8 d), 4 d in 15 mM glucose, 8 d in 15 mM glucose, 4 d in 30 mM glucose, 8 d in 30 mM glucose, memory 4 d + 4 d (4 d in 30 mM glucose followed by 4 d in 5.5 mM glucose), memory 4 d + 8 d (4 d in 30 mM glucose followed by 8 d in 5.5 mM glucose). Three biological replicates, each with three technical replicates, were used per treatment. Differences between groups were examined using one-way ANOVA, and then, pairwise comparisons were made using Tukey's test. *$P < 0.05$ versus control, **$P < 0.01$ versus control, ***$P < 0.001$ versus control. **(C)** Percentage of cells in each cell cycle phase evaluated by flow cytometry and DAPI staining. Three biological replicates per treatment were used. Differences between groups were tested using one-way ANOVA, and then, pairwise comparisons were made using Tukey's test. *$P < 0.05$ versus control. **(D)** Flow cytometry assays measuring relative ROS concentrations by dihydrorhodamine 123 and MitoSOX, and relative NO concentrations using DAF-2 DA. Three biological replicates were used per treatment. Differences between groups were examined with one-way ANOVA, and then, pairwise comparisons were made using Tukey's test.

osmolarity control. We found that neither the extracellular acidification rate (ECAR)—measurement indicative of glycolysis—nor the oxygen consumption rate (OCR) was altered in response to our treatments (Fig S1A). To ensure that glucose treatments were not inhibiting cell proliferation, we measured cell growth in our cultures and found that HG exposure had a marginal effect on cell division on days 7 and 8 of culture (Fig S1B). Nonetheless, the rate of cell division was maintained at ~1 cell division every 24 h across all treatments (Fig S1C). In line with this, cell cycle assessment showed that only HG treatment had differences in the proportion of cells in each phase compared with the control (Fig 1C): HG-treated cells had an increase of 3.7% of cells in the G2/M phase concomitant with a decrease of 5.1% of cells in the S phase, indicative of a marginal delay in cell cycle. In addition, we did not detect differences in cell

viability between our treatments as measured by a calcein-AM assay (Fig S1D).

All of the above supports the existence of an in vitro transcriptional memory that is inherited through cell divisions under our experimental conditions. In addition, hyperglycemia has also been shown to induce persistent oxidative stress, so we performed assays with two different molecules, dihydrorhodamine 123, which predominantly detects peroxynitrite (Haddad et al, 1994; Crow, 1997), and MitoSOX, which predominantly detects superoxide (Robinson et al, 2006). Our results show that HG induced an increase in ROS measured by dihydrorhodamine 123 and MitoSOX, which persisted in the memory treatment (Fig 1D); consistent with previous studies (Paneni et al, 2012; Yao et al, 2022). On the contrary, as nitric oxide (NO) production

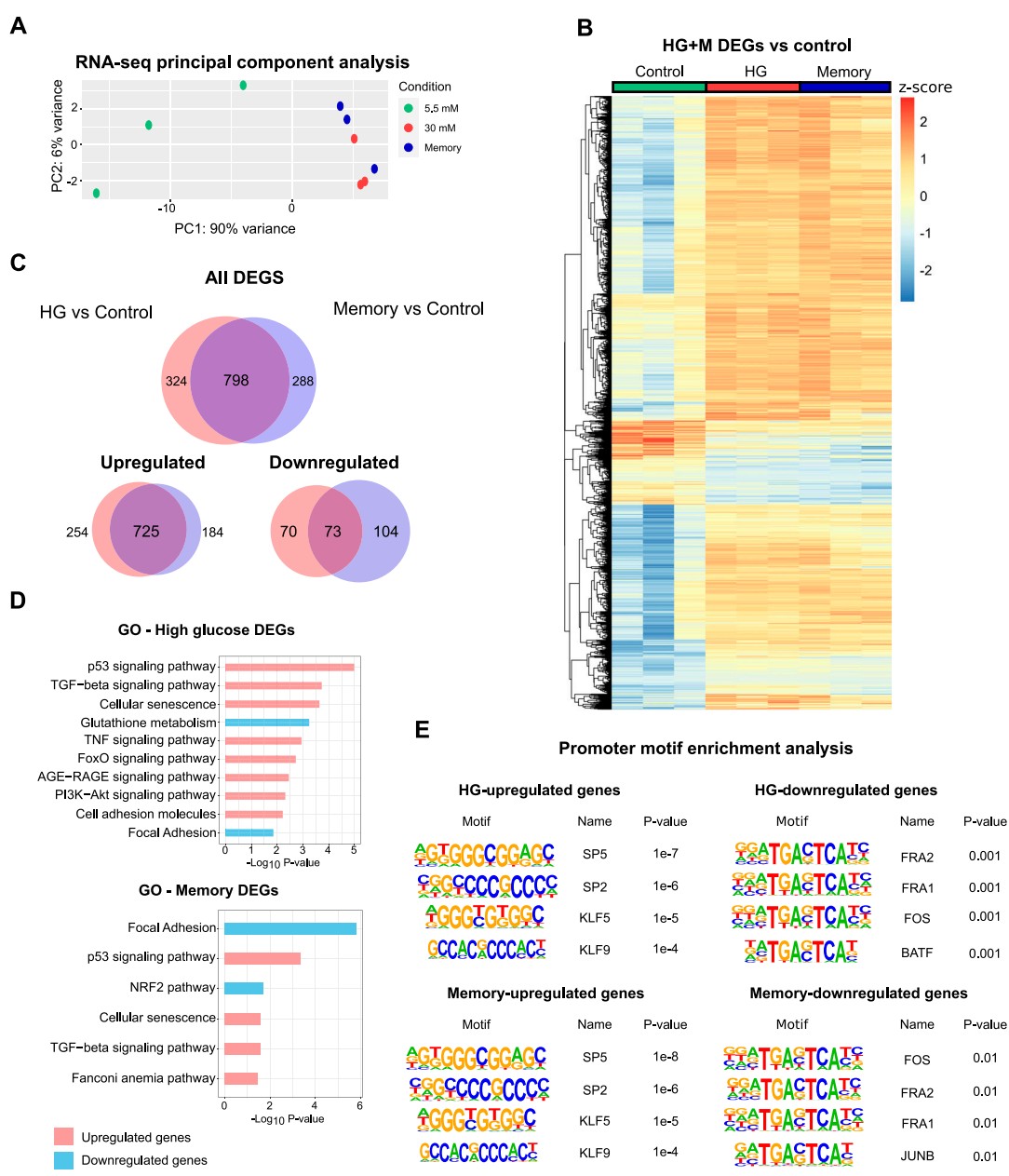

**Figure 2. High-glucose–induced transcriptional memory in endothelial cells.**
**(A)** Principal component analysis of RNA-seq samples with three biological replicates per treatment. **(B)** Heatmap displaying the relative expression changes of 1,410 differentially expressed genes (DEGs): HG versus control, and memory versus control. **(C)** Venn diagram showing the overlap between HG versus control, and memory versus control DEGs. **(D)** Pathway enrichment analysis of HG versus control, and memory versus control DEGs. **(E)** Transcription factor motif enrichment analysis conducted in promoters of HG versus control, and memory versus control DEGs. A window ranging from −1,000 to +100 bp from the TSS was used for the search.

is impaired in the vascular endothelium in diabetes (Kolluru et al, 2012), we evaluated NO concentrations using DAF-2 DA (Berkels et al, 2000), finding no differences induced by our experimental treatments compared with the control. Although other methodologies to quantify NO production could be implemented, these results are indicative that transient hyperglycemia induces an oxidative memory in human ECs.

To further characterize the persistent effects of transient HG in ECs, we analyzed the cell transcriptome across treatments.

Principal component analysis of the transcriptomic profiles showed that cells exposed to HG and memory treatments were highly similar (Figs 2A and S2A). Consistent with this, only 16 differentially expressed genes (DEGs) were found between HG and memory treatments (Fig S2B; adjusted $P < 0.05$ and $\log_2$ fold change $> 0.5$ and $< -0.5$; DEGs of pairwise comparisons can be found in Supplemental Data 1). Conversely, in the differential gene expression analysis against the control, we found 1,122 DEGs in HG and 1,086 DEGs in memory (M) treatment (Fig 2B and C). Of these, 798 (71% of

HG DEGs and 73% of memory DEGs) were common between HG and memory (henceforth referred to as HG/M-shared DEGs). These results support the existence of a transcriptional memory induced by transient high glucose. We next used the HG DEGs and memory DEGs to conduct gene set enrichment analyses to identify over-represented pathways associated with these experimental conditions. First, we independently examined HG and memory DEGs and found that up-regulated HG DEGs were enriched for terms and pathways known to be affected by hyperglycemia and diabetes(Yano et al, 2004; Coughlan et al, 2009; Deshpande et al, 2013; Zhang et al, 2015; Kyriazis et al, 2021), such as the p53, TGF-$\beta$, PI3K-Akt, TNF, FoxO, and AGE-RAGE pathways, as well as cellular senescence and adhesion molecules (Fig 2D). Moreover, down-regulated HG DEGs highlighted terms such as glutathione metabolism and focal adhesion. Similarly to the case of HG DEGs, up-regulated memory DEGs were enriched for terms such as cellular senescence, and p53 and TGF-$\beta$ pathways, whereas down-regulated memory DEGs showed terms such as focal adhesion and the PI3K-Akt and NRF2 signaling pathways (Fig 2D). Many of these pathways were confirmed using the GSEA (Subramanian et al, 2005) software analysis (Fig S2C).

We next examined the 798 HG/M-shared DEGs, and we found that up-regulated HG/M-shared DEGs were enriched for terms such as p53 and TGF-$\beta$ pathways (Fig S3A). Furthermore, down-regulated HG/M-shared DEGs were enriched for terms of focal adhesion and the PI3K-Akt signaling pathway. To evaluate the distinctive transcriptomic profile of transient hyperglycemia exposure, we examined independently both the 324 DEGs unique to HG and the 288 DEGs exclusive to the memory treatment ("HG-unique" and "memory-unique," respectively). Interestingly, up-regulated HG-unique genes were enriched for terms such as the TNF, TGF-$\beta$, and AGE-RAGE pathway, as well as cellular senescence and adhesion molecules (Fig S3C). In contrast, down-regulated HG-unique genes were enriched for glutathione metabolism. On the contrary, up-regulated memory-unique genes were enriched for the Fanconi anemia pathway, linked to DNA repair and replication, whereas memory-unique down-regulated genes were enriched for focal adhesion and the NRF2 pathway (Fig S3C).

To gain insights into the transcriptional regulatory program underlying the glucose-induced memory, we conducted analyses of possible transcription factors (TFs) driving persistent transcriptional changes. To this end, we examined the promoter sequence of HG and memory DEGs to perform a TF motif enrichment query. Our analysis revealed that the KLF/SP family of transcription factors exhibited the most statistically enriched motifs in the up-regulated genes of both HG and memory (Fig 2E). This finding aligns with our differential expression analysis, which identified KLF3, KLF5, KLF9, and SP4 as persistently up-regulated in the memory treatment (Fig S4C). In contrast, the promoters of down-regulated HG and memory genes were enriched for the predicted motifs of FRA1, FOS, and JUN (Fig 2E). Of note, FRA1 (*FOSL1*) was down-regulated in HG and memory treatments (Fig S4C). We noticed that these motifs share the same core sequence "TGA(C/G)TCA," which corresponds to the common binding motif of the bZIP domain transcription factors (Rodríguez-Martínez et al, 2017). This family includes the transcription factor NRF2, known to be the master regulator of the antioxidant response in mammals. Collectively, these findings

suggest that transient high glucose triggers a persistent transcriptional memory in human ECs, with bZIP transcription factors potentially playing an important role.

## Transient high-glucose treatment induces persistent chromatin accessibility changes in non-promoter regions

To further investigate the mechanisms underlying glucose-induced transcriptional memory, we performed ATAC-seq to assess potential chromatin accessibility changes associated with the HG and memory treatments. We identified 79,376 unique peaks, with 45,363 shared across treatments (Fig S5A). The distribution of called peaks was consistent across conditions, with the majority located in introns, followed by intergenic and promoter regions (Fig S5B). Analysis of the ATAC-seq signal around peaks by genomic feature revealed an increase in accessibility in intergenic and intronic regions in HG and memory, which was absent in peaks at promoters or exons (Fig S5C). Differential chromatin accessibility analysis of HG versus control identified 2,863 differentially accessible regions (DARs; fold change >1.5 and $P < 0.0001$); over 90% of which were located in either intronic or intergenic regions (Fig 3A). Among these DARs, 2,641 (92%) had increased accessibility in HG versus control (Fig 3C). Similarly, in the memory versus control comparison, 1,869 DARs were identified, with 89% of them located in either intronic or intergenic regions (Fig 3A). In addition, most of the memory DARs (81%) had increased accessibility compared with the control (Fig 3C). Notably, inspection of the mean ATAC signal across the 4,166 HG plus memory DARs revealed that changes in these regions were not fully reverted to control levels in the memory treatment (Fig 3C), indicating the establishment of an epigenetic memory induced by transient hyperglycemia.

Given that most of the identified DARs were in intergenic and intronic regions, and considering that genomic long-range interactions play critical roles in fine-tuning gene expression in response to the environment (Maurya, 2021), we explored whether non-promoter DARs could be annotated to enhancer regions. Using HUVEC ChIP-seq data from ENCODE for two enhancer-enriched histone marks, H3K4me1 and H3K27ac, we found 869 and 487 DARs enriched with both histone marks in HG and memory, respectively, so we catalogued these regions as differentially accessible putative enhancers (DAEs; Fig 3B; Supplemental Data 2). Next, as three-dimensional genome organization has been shown to play an important role in regulating and delimiting enhancer–promoter contacts (Yang & Hansen, 2024), we employed publicly available HUVEC high-resolution Hi-C data to identify the potential target genes of our identified DAEs (Rao et al, 2014), where only the genes located within the same topological domain were considered putative target genes of a given enhancer (Fig 3D). Remarkably, we found 132 HG DAEs and 73 M DAEs whose regulatory domain contained at least one HG+M DEG in the shared Hi-C domain. When we examined the correlation between changes in accessibility and gene expression, we observed that 107 (81%) and 57 (78%) of HG DAEs and memory DAEs, respectively, changed in the same direction as their associated DEG; these genes were classified as putative targets of our DAEs (Fig 3E; Supplemental Data 2).

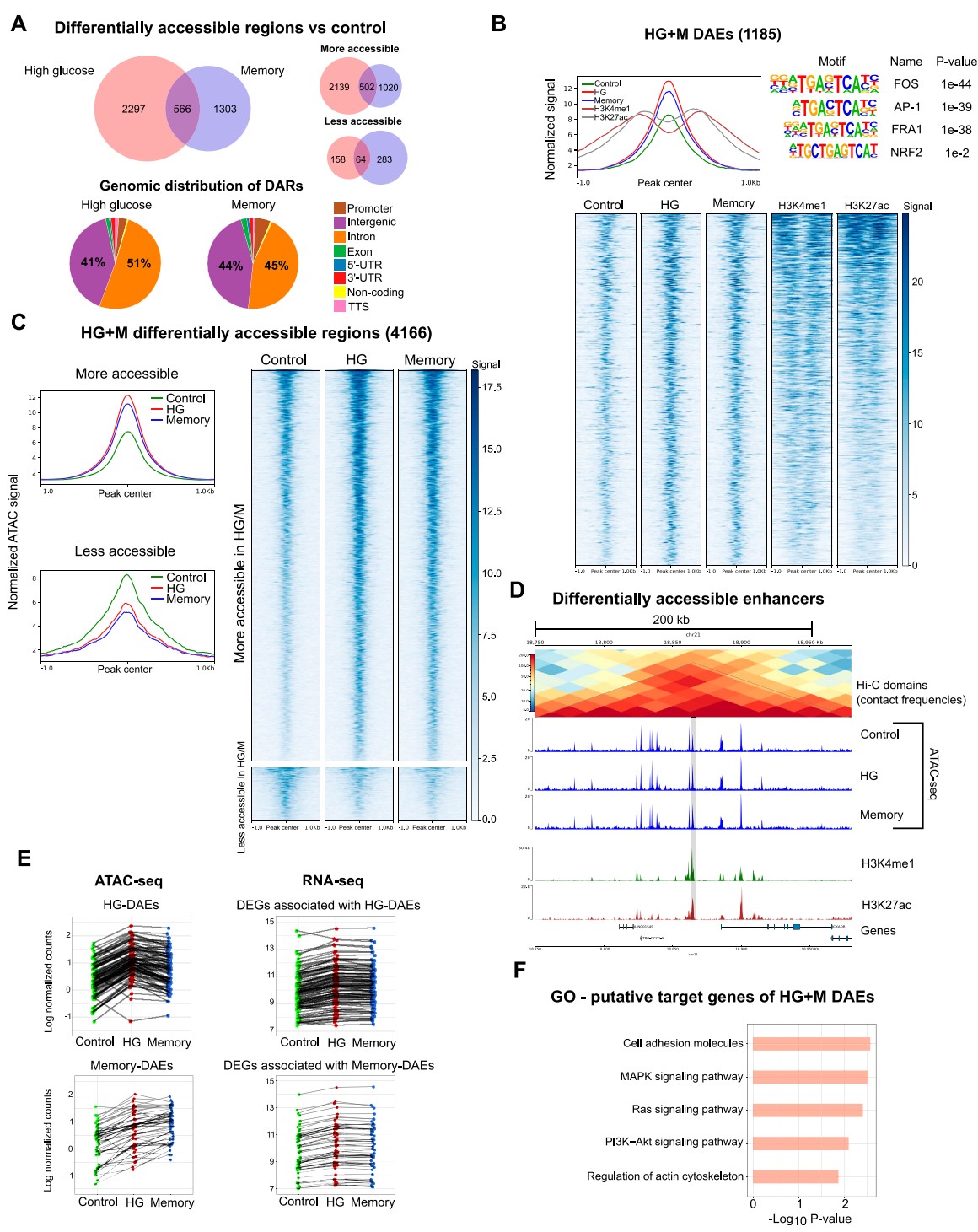

**Figure 3. Transient high glucose induces persistent changes in HUVEC chromatin accessibility.**
**(A)** Top: Venn diagrams showing the overlap between high glucose (HG) versus control, and memory (M) versus control differentially accessible regions (DARs). Bottom: pie charts showing the genomic distribution of DARs. **(B)** Upper left: ATAC-seq signal plot showing the control, HG, and memory, alongside H3K4me1 and H3K27ac ChIP-seq signal in differentially accessible enhancers (DAEs). Upper right: transcription factor motif enrichment analysis in DAEs within ±50 bp from the ATAC-seq peak center. Bottom: heatmap of DAEs showing the control, HG, and memory ATAC-seq signal along with the H3K4me1 and H3K27ac ChIP-seq signal. **(C)** Left: normalized ATAC-seq signal of more accessible and less accessible HG + memory DARs in control, HG, and memory samples. Right: heatmap of ATAC-seq signal in HG + M DARs in control, HG, and memory samples. **(D)** Example of a differentially accessible enhancer (highlighted in gray) located inside a genomic topological domain; note that chromatin accessibility is not reverted to the control levels in the memory treatment. **(E)** Association between changes in chromatin accessibility in HG and memory DAEs and changes in the gene expression of putative target DEGs identified employing HUVEC Hi-C data. **(F)** Pathway enrichment analysis of the HG + M DEGs that are putative targets of DAEs.

In addition, pathway analysis of these putative target genes highlighted terms such as cell adhesion molecules, MAPK signaling pathway, Ras signaling pathway, and PI3K-Akt signaling pathway (Fig 3F). Finally, to identify candidate TFs driving the glucose-induced chromatin accessibility changes, we searched for enriched motifs within HG + M DAEs. We found that DAEs were significantly enriched for motifs of bZIP transcription factors, including FOS, AP-1, FRA1, and NRF2 (Fig 3B). These findings are consistent with our earlier motif enrichment analysis of down-regulated HG and memory genes that suggested that bZIP transcription factors influence the transcriptional response in HG and memory.

In summary, we found that transient hyperglycemia induces persistent chromatin accessibility changes mainly in non-promoter regions, including putative enhancers, which could contribute to the glucose-induced transcriptional changes. Furthermore, bZIP transcription factors emerge once more as potential regulators of these processes.

## Activation of the NRF2 pathway reverts the glucose-induced transcriptional and epigenetic memories

Our findings indicated that HG and memory treatments presented down-regulation of antioxidant response-related genes, including those in the glutathione metabolism and NRF2 pathway. In addition, we observed bZIP motif enrichment at promoters of down-regulated memory DEGs and HG + M differentially accessible enhancers. These prompted us to explore the role of the transcription factor NRF2 in the persistent effects of transient high glucose in ECs. We first used lentiviral vectors to express GFP (as a control) or human NRF2 in HUVECs, and then exposed these cells to the HG and memory treatments to evaluate transcriptional changes via RNA-seq (Fig 4A). To confirm NRF2 overexpression (OE), we validated its expression in our transduced RNA-seq samples, finding a >10-fold increase in transcript abundance compared with the control (Fig S4C). Then, we evaluated the effect of NRF2 OE during HG and memory treatments by analyzing changes in the expression of their corresponding subsets of DEGs. We found that only 26% (281 of 1,086) of the memory DEGs, and 20% (220 of 1,122) of the HG DEGs remained differentially expressed compared with the control after NRF2 OE (Fig 4B and C). These results indicate that NRF2 OE effectively mitigates the transcriptional effects of HG exposure. Subsequently, we performed pathway enrichment analysis on both NRF2 OE–restored and not-restored genes. Interestingly, NRF2 OE–restored genes were enriched for terms related to cellular senescence, Fanconi anemia, focal adhesion, and glutathione metabolism, as well as the p53 and FoxO pathways (Fig 4E). Meanwhile, NRF2 OE–not-restored genes showed terms related to cellular senescence, adhesion molecules, and the TGF-β, p53, and NF-κB pathways, indicative of NRF2 OE being incapable of reverting changes in pro-inflammatory genes (Fig S3D).

To circumvent the disadvantages of lentiviral systems for therapeutic purposes, we used sulforaphane (SF) to induce the NRF2 pathway. SF exerts NRF2 activation by inhibiting its interaction with the repressor KEAP1, leading to NRF2 accumulation and nuclear translocation. We tested various SF concentrations and determined that 1 $\mu$M effectively induced the expression of NRF2 target genes HMOX1 and NQO1 without impacting cell viability or

proliferation (Figs S1B–D and S4B). In agreement with the mechanism of NRF2 activation by SF, culture of HUVECs with 1 $\mu$M SF did not induce NRF2 transcription changes (Fig S4B). Moreover, RT–qPCR of selected HG/M DEGs confirmed that 1 $\mu$M SF reverted their expression changes in our HG and memory treatments (Fig S4A); hence, we used this concentration for subsequent experiments.

We performed RNA-seq on cells exposed to our HG and memory treatments, by adding SF throughout the entire HG treatment, or after switching to normal glucose in the case of the memory treatment (Fig 4A). This allowed us to evaluate whether SF could prevent or reverse the glucose-induced transcriptional memory. Remarkably, SF ameliorated the transcriptional alterations in both the HG and memory treatments, where only 471 of 1,086 (43%) memory DEGs and 572 of 1,122 (51%) HG DEGs remained differentially expressed after SF treatment (Fig 4B–D). Interestingly, upon examining the SF-restored genes, we found notable differences in the enriched pathways compared with those of NRF2 OE–restored genes: SF-restored genes highlighted terms such as the TGF-β signaling pathway and TNF signaling pathway, which were absent in the NRF2 OE–restored genes (Fig 4E and F). Additional terms enriched in the SF-restored genes included cytokine–cytokine receptor interaction, and the PI3K-Akt, FoxO, and NRF2 pathways, as well as focal adhesion and glutathione metabolism. On the contrary, genes not restored by SF supplementation showed terms such as cellular senescence and the p53 and Fanconi anemia pathways (Fig S3B). The effect of NRF2 activation, either by NRF2 OE or by SF treatment, on the expression of DEGs belonging to selected pathways is shown in Fig 4D.

In addition to evaluating the effect of NRF2 activation at the transcriptional level, we also performed ATAC-seq in HUVECs exposed to our memory + SF treatment to assess its potential beneficial effects in reverting the persistent high-glucose–induced chromatin accessibility changes. Strikingly, our findings revealed that SF restored the accessibility in 1,489 (80%) of the 1,869 memory DARs (Fig 5A–C). Moreover, among the 30 memory DAEs associated with memory DEGs, 15 were restored by SF supplementation, and when we examined the association between SF-mediated reversal of chromatin accessibility changes and reversal of gene expression changes in these 15 putative enhancer–gene pairs, we found that 12 of them had both chromatin accessibility and gene expression restored by SF (Fig 5D). These results suggest that SF may counteract HG-induced gene expression alterations by reinstating the epigenetic landscape at their associated enhancers.

Finally, we evaluated the effect of SF supplementation in ROS concentrations in HUVECs exposed to our treatments using dihydrorhodamine 123 (DHR 123) and MitoSOX assays. Our results revealed that in the case of DHR 123, SF reduced ROS concentrations in both HG + SF and memory + SF compared with their respective unsupplemented condition; meanwhile, this reduction was observable only in the HG + SF treatment in the case of the MitoSOX assay (Fig 5E). This discrepancy suggests potential variations in the production of different ROS after transient HG exposure, and/or differences in the mechanism by which SF elicits its effects.

Taken together, our results show that NRF2 activation effectively reverts the high-glucose–induced transcriptional and epigenetic memories, while also mitigating the persistent increase in ROS after transient high-glucose exposure. These findings suggest that

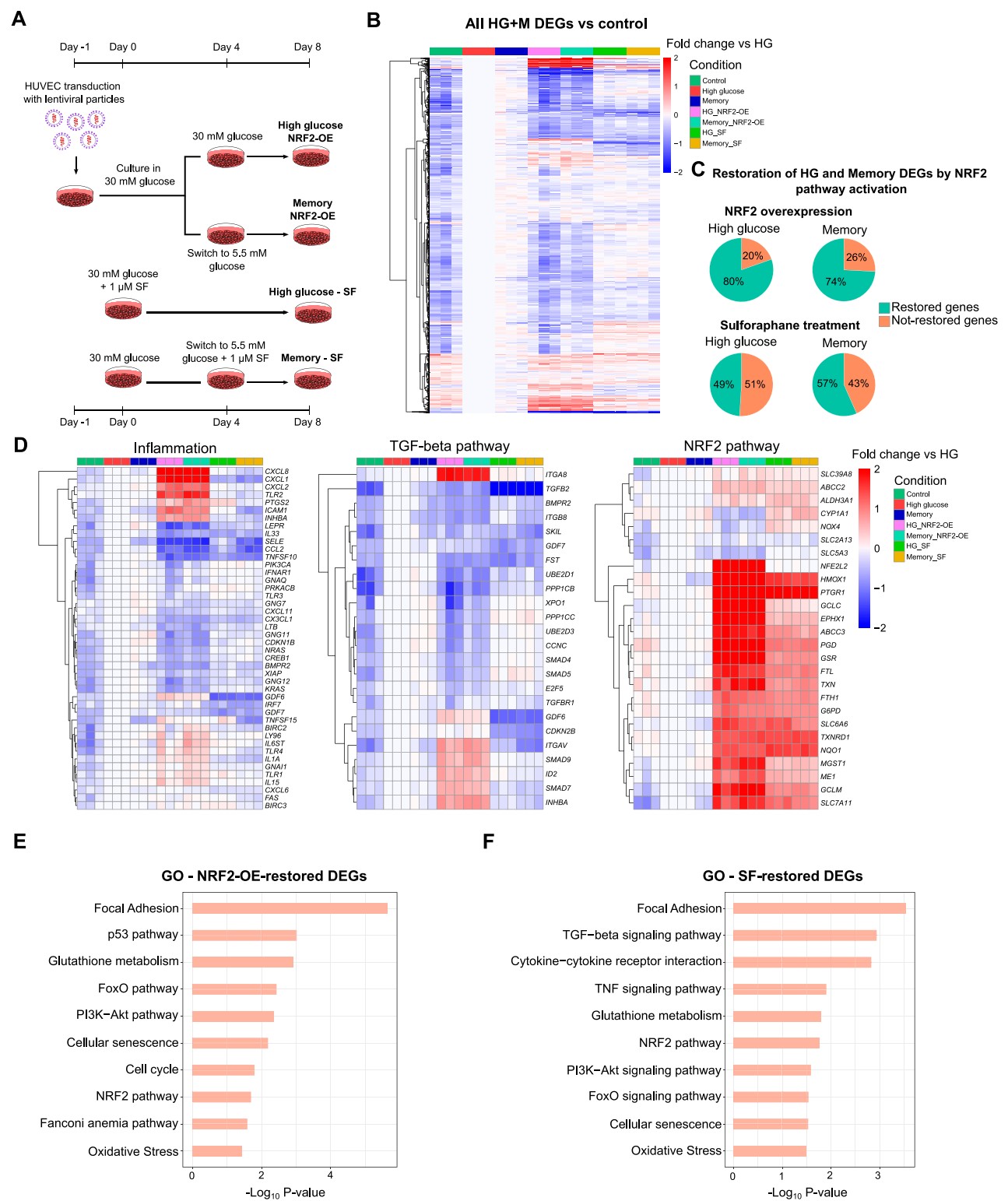

**Figure 4. NRF2 pathway activation via gene overexpression or sulforaphane supplementation reverts high-glucose–induced transcriptional changes.**
**(A)** Protocol used for NRF2 overexpression (OE) or sulforaphane (SF) supplementation in HUVECs. For NRF2 OE, cells were transduced with NRF2-carrying lentiviral particles (MOI = 20) while cultured in 5.5 mM glucose. After 24 h, glucose concentration was increased to 30 mM for 8 d (HG treatment), or 4 d in 30 mM followed by 4 d in 5.5 mM of glucose for memory treatment. For SF treatments, 1 μM SF was added during the 8-d HG treatment or at the switch time point of the memory treatment (final 4 d in 5.5 mM glucose). **(B)** Heatmap depicting the effect of NRF2 activation by OE or SF treatment on HG + M DEGs. Data are shown as a fold change versus HG. **(C)** Percentage of DEGs found in HG or memory treatments that are restored via NRF2 pathway activation by NRF2 OE or SF supplementation. **(D)** Heatmaps of HG/M DEGs grouped in three categories: inflammation, TGF-β pathway, and NRF2 pathway. Data are shown as a fold change versus HG. **(E)** Pathway enrichment analysis of HG + M DEGs restored by NRF2 OE supplementation. **(F)** Pathway enrichment analysis of HG + M DEGs restored by sulforaphane supplementation.

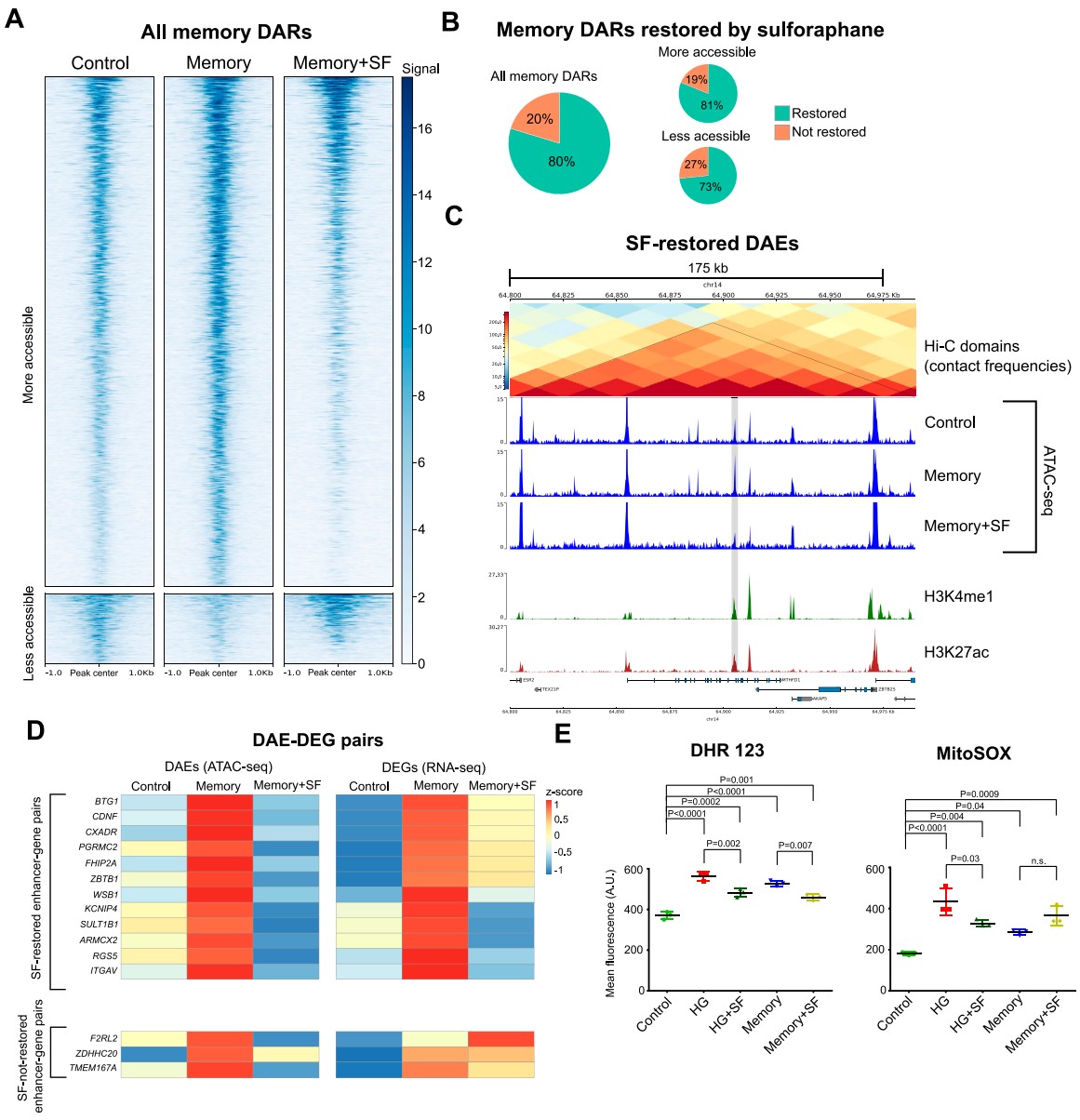

**Figure 5. Sulforaphane supplementation reverts high-glucose–induced chromatin accessibility changes.**
**(A)** Heatmap showing the ATAC-seq signal in memory DARs in control, memory, and memory + sulforaphane (SF). **(B)** Proportion of memory DARs restored by SF supplementation. **(C)** Example of a memory DAE restored by SF supplementation (highlighted in gray). **(D)** Heatmap showing the association between restoration of chromatin accessibility (left) in DAEs after SF treatment, and restoration in the gene expression (right) of their respective putative target DEGs. The names of the genes are depicted on the left. Note that most of DAEs restored by SF had their putative target gene restored. **(E)** Flow cytometry assays measuring relative ROS concentrations by dihydrorhodamine 123 and MitoSOX. Three biological replicates were used per treatment. Differences between groups were examined with one-way ANOVA, and then, pairwise comparisons were made using Tukey's test.

modulating the NRF2 pathway could be a promising therapeutic approach to counteract the enduring effects induced by hyperglycemia.

# Discussion

Transient pathological hyperglycemia can alter vascular homeostasis by inducing sustained changes in gene expression, such as up-regulation of pro-inflammatory pathways and increased oxidative stress (Brasacchio et al, 2009; Paneni et al, 2012). Here, we report that activation of the NRF2 pathway, by genetic overexpression or pharmacological activation with sulforaphane, can revert the transcriptional and epigenetic memories induced by high glucose in human ECs. Remarkably, sulforaphane not only prevents most of the aforementioned alterations caused by HG, but it can also revert them once established. Sulforaphane is a plant-derived isothiocyanate found in cruciferous vegetables, possessing diverse

biological effects (Mangla et al, 2021). For instance, sulforaphane blocks inflammation by inhibiting NF-κB activity, and has antioxidant properties by modifying cysteine residues of KEAP1, the inhibitor of NRF2, resulting in NRF2 activation and subsequent induction of antioxidant and stress response genes (Hu et al, 2011). Although NRF2-independent chemoprotective mechanisms for sulforaphane have been described (Greaney et al, 2016), our data showing that NRF2 gene overexpression resulted in a similar outcome suggest that the beneficial effects conferred by sulforaphane in our HG and memory treatments occur mainly through the activation of the NRF2 pathway.

Interestingly, although NRF2 OE restored the expression of more DEGs in either HG or memory compared with sulforaphane treatment, our RNA-seq analysis revealed that NRF2 OE led to the increased expression of pro-inflammatory genes, surpassing even the levels observed in HG/M alone in some cases (see "inflammation" in Fig 4D). We hypothesize that this effect resulted from the lentiviral-mediated overexpression of NRF2 beyond the physiological range. In line with this, previous studies reported that NRF2 can activate pro-inflammatory genes in some contexts (Wruck et al, 2011). In fact, NRF2 ablation is reported to decrease NF-κB subunits p50 and p65 in mouse fibroblasts (Yang et al, 2005), highlighting the importance of TF dosage in modulating its function and binding to target genes.

Many unresolved questions about the metabolic memory revolve around the molecular basis driving the establishment and persistence of glucose-induced transcriptional memory. Epigenetic regulation of gene expression stands out as a possible mechanism for the modulation of these processes (Reddy et al, 2015; Dhawan et al, 2022). With that in mind, we assessed the chromatin accessibility landscape of HUVECs exposed to our HG and memory treatments, finding that hyperglycemia led to changes in accessibility specifically within intergenic and intronic regions that are not reversed post-glucose normalization. Interestingly, most of these DARs resulted from an increase in accessibility, and when we overlapped them with HUVEC H3K4me1 and H3K27ac ChIP-seq data, we found that a proportion of those regions were *bona fide* enhancers. Moreover, using HUVEC Hi-C data, we identified differentially accessible enhancers that could potentially co-regulate some of the genes found to be altered in our HG and memory treatments. Notably, sulforaphane supplementation restored a substantial proportion of the regions that were differentially accessible after transient high-glucose exposure. In addition, we observed that most of the putative enhancers that were restored by SF had the expression of their associated DEG restored, suggesting that enhancer dynamics contribute to the glucose-induced transcriptional alterations, and that one of the mechanisms of SF-mediated reversal of the transcriptional memory is through reversal of epigenetic changes in genome regulatory regions, such as enhancers. Our observation that enhancers retain an accessibility memory is relevant, given their role in controlling the magnitude and timing of gene expression (Maurya, 2021). Based on their epigenetic signatures, enhancers can reside in "active," "inactive," or "poised" states (Creyghton et al, 2010). Thus, we hypothesize that transient hyperglycemia impacts the epigenetic and functional state of enhancers, priming them

to amplify or sustain the transcriptional changes. This mechanism mirrors how inflammation can imprint an enhancer's epigenetic memory in immune cells and ECs (Drummer et al, 2021). Ergo, in diabetes patients, repetitive cycles of pathological hyperglycemia could set enhancers into a pathological memory state.

Additionally, we found differential enrichment of TF binding motifs between promoters of HG/M up-regulated and down-regulated genes. Specifically, up-regulated genes were enriched in motifs of the KLF/SP family of TFs, whereas down-regulated genes favored members of the bZIP family. Concordantly, we observed up-regulation of KLF3, KLF5, KLF9, and SP4 in HG and memory treatments, and notably, all of these genes were restored by sulforaphane treatment, suggesting that one possible mechanism by which NRF2 reverts the glucose-induced transcriptional memory is through the restoration of the expression of these transcription factors. Interestingly, elevated KLF5 expression has been linked to oxidative stress, vascular remodeling, and impaired lipid metabolism (Kyriazis et al, 2021).

Meanwhile, promoters of down-regulated HG/M DEGs exhibited enrichment for bZIP family members, which share common features in their binding motifs and genomic targets (Rodríguez-Martínez et al, 2017). Notably, we identified the bZIP member FRA1 consistently down-regulated in our HG and memory treatments, and similar to KLF/SP TFs, FRA1 expression was restored after sulforaphane treatment. FRA1 is a member of the AP-1 complex known to induce the expression of antioxidant genes such as HMOX1 and sulfiredoxin (Lee et al, 2000; Soriano et al, 2009)—genes persistently down-regulated in our HG and memory treatments and modulated by NRF2. This observation, together with previous reports showing that NRF2 protein levels are persistently diminished after transient HG (Yao et al, 2022), could explain the paradoxical finding in various studies (Hodgkinson et al, 2003; Kowluru et al, 2007; Sekhar et al, 2011), including our own, regarding the decreased expression of antioxidant genes under HG. Nonetheless, research on the interplay between these TFs regarding their activities and shared molecular targets is still lacking.

Overall, our work demonstrates that transient hyperglycemia induces prolonged alterations in the transcriptome, oxidative stress, and chromatin accessibility in cultured ECs, but some limitations should be considered. First, in our study, ECs underwent four divisions after the hyperglycemic stimulus, yet the cell division rate in adults is slower and influenced by age (Hobson & Denekamp, 1984; Hoshi & McKeehan, 1986). Hence, it is difficult to extrapolate the time required to establish a glucose-induced metabolic memory in the adult vasculature. Second, in vivo, the vasculature is subjected to multiple stimuli, such as shear stress from blood flow, and exposure to glucose fluctuations throughout the day (Zhou et al, 2020), including exposure to lower pathological concentrations, such as 15 mM, which in our model did not induce a robust memory effect. Third, our results revealed that sulforaphane could reverse the transcriptional and epigenetic changes associated with the glucose-induced memory. Still, it would be relevant to assess—especially in vivo—whether this reversal is accompanied by metabolic and epigenetic changes, in particular, histone modifications

and 3D chromatin architecture. Fourth, although we verified that NRF2 pathway activation can prevent and revert the transcriptional changes caused by transient HG, our data are insufficient to determine whether this effect results from direct action of this TF and/or from downstream regulation of NRF2 targets. Likewise, the impact of activating NRF2 in normoglycemia warrants investigation. Lastly, our study focused only on HUVECs; thus, other physiologically relevant ECs or in vivo models would have to be investigated.

The metabolic memory phenomenon has been studied for over three decades (Lachin et al, 2021), yet currently, there are no specific treatments to ameliorate the diabetes-associated vascular complications, which comprise the leading causes of morbidity and mortality in patients with this disease. Our study highlights the potential use of sulforaphane to revert the high-glucose–induced transcriptional and epigenetic memories in human ECs. Small-scale studies using sulforaphane in diabetes patients have shown promising results (Axelsson et al, 2017). Nonetheless, we look forward to future randomized longitudinal studies with larger cohorts that will assess the efficacy and safety of long-term sulforaphane supplementation in mitigating the impact of diabetes-associated metabolic memory.

# Materials and Methods

### Cell culture

Pooled-donor HUVECs (C2519A, Lot. No. 0000478982, 18TL169251, and 23T2043669; Lonza) were cultured in endothelial basal medium (EBM; CC-3121; Lonza) supplemented with SingleQuots (CC-4133; Lonza) at 37°C, 5% $CO_2$, and 8% $O_2$ for no more than eight passages, using 0.05% trypsin to subculture the cells. HUVECs were seeded in precoated gelatine plates; the medium was replaced every 48 h. Cells were cultured for 8 d in glucose 5.5 mM (control), glucose 30 mM (high glucose), or memory treatment (4 d at 30 mM, then 4 d at 5.5 mM). The seeding number was adjusted to reach 70–80% confluency after 4 d of culture, and only one passage was performed throughout the 8 d of treatment (at day 4). 1 $\mu$M sulforaphane (S4441; Sigma-Aldrich) was added into media from a 2.8-mM stock. Supplementation was maintained for 8 d in 5.5 or 30 mM glucose, or when cells switched to 5.5 mM in the memory treatment. Controls received an equivalent volume of vehicle (DMSO). For some experiments, we added mannitol, a non-metabolizable sugar, to a final concentration of 25 mM to evaluate potential changes induced by osmolarity. Biological replicates in our experiments came from cells cultured in different wells and/or days.

### Cell viability assays

For calcein AM (C3100MP; Thermo Fisher Scientific), cells were harvested after treatments and washed with PBS, then resuspended in HBSS with 1 $\mu$M calcein AM for 20 min at 37°C, protected from light. The dye was then removed, and fresh HBSS was added. Fluorescence intensity was measured with FACSMelody Cell Sorter (BD). Single cells were gated, and 10,000 cells were acquired per sample to identify high-fluorescent viable cells. Unstained cells were used as controls. Data were analyzed with FlowJo. Three biological replicates were acquired from each treatment.

### Cell growth curves and cell cycle

HUVECs were seeded in gelatin-coated 12-well plates and counted daily for 8 d using the CytoSMART counter (Corning). Three biological replicates with two technical replicates each were evaluated per day per treatment. At day 4, cells were subcultured. To maintain a consistent curve, cell counts from days 5–8 were calculated using a division rate derived from day 4 cell counts. This rate was the day's total cells divided by starting cells from day 4 subculture.

To obtain the overall mean division rate, we used the following formula:

$$\frac{\log_2(final\ number\ of\ cells/starting\ number\ of\ cells)}{time(days)}$$

For cell cycle analysis, cells were trypsinized and resuspended in DPBS with 0.1% NP-40 and DAPI (1 $\mu$g/ml). After 10-min incubation, cell DNA content was measured with FACSMelody Cell Sorter (BD) using a UV laser on a linear scale. 5,000–10,000 single cells were acquired per sample with three biological replicates per treatment. FlowJo software was used to calculate the proportion of cells in each cell cycle phase.

### ROS and NO assays

For the DHR 123 assays, cells were trypsinized, washed with DPBS, and resuspended in HBSS with 1 $\mu$g/ml DHR 123 (Thermo Fisher Scientific), protected from light. After 20 min at room temperature, green (488/527) fluorescence was directly evaluated with FACS-Melody Cell Sorter (BD). 5,000–10,000 single cells were acquired per sample with three biological replicates per treatment. Unstained cells served as controls. Analysis was done using FlowJo v10, with mean fluorescence indicating ROS levels.

For the MitoSOX assays, cells were trypsinized, washed with DPBS, and resuspended in HBSS with 2.5 $\mu$M MitoSOX Red (Thermo Fisher Scientific), and then, cells were incubated for 30 min at 37°C, protected from light. After the incubation, cells were rinsed twice with DPBS and resuspended in HBSS. Fluorescence (488/613) was measured with FACSMelody Cell Sorter (BD). 5,000–10,000 single cells were acquired per sample with three biological replicates per treatment. Analysis was done using FlowJo v10, with mean fluorescence indicating ROS levels.

For the NO assays, cells were trypsinized and washed with DPBS, and then, they were resuspended in HBSS with 10 $\mu$M DAF-2 DA (Calbiochem). After incubation for 30 min at 37°C, protected from light, cells were rinsed twice with DPBS and resuspended in HBSS. Fluorescence (488/527) was measured with FACSMelody Cell Sorter (BD). 5,000–10,000 single cells were acquired per sample with three biological replicates per treatment. Analysis was done using FlowJo software, with mean fluorescence indicating NO levels.

### Extracellular flux assays

ECAR and OCR were measured using the Seahorse XFe96 analyzer (Agilent) with Cell Energy Phenotype Test Kit. The day before, 10,000 HUVECs/well from each treatment were seeded in a gelatin-precoated 96-well plate and cultured in EBM with the corresponding glucose concentration at 37°C, 5% $CO_2$, and 8% $O_2$. On assay day, cells were washed with 200 $\mu$l assay medium (Seahorse XF Base Medium with 1 mM pyruvate, 2 mM glutamine, and 10 mM glucose) and incubated in 180 $\mu$l of fresh assay medium for 1 h at 37°C before the assay. The Cell Energy Phenotype Test consists of three baseline measurements of OCR and ECAR followed by five stressed measurements. The latter are induced by the simultaneous addition of oligomycin, a complex V inhibitor, and carbonyl cyanide-p-trifluoromethoxyphenylhydrazone (FCCP), an uncoupling agent. Oligomycin and FCCP were added to each well to a final concentration of 1 mM and 0.5 mM, respectively. Results were analyzed with Agilent's Wave software and normalized to cell number. The cell number of each well was determined by adding Hoechst to a final concentration of 2 $\mu$M. Then, nuclei were counted using BioTek Cytation 1 Cell Imaging Multimode Reader. Nine biological replicates were used per condition, each one with multiple technical replicates.

### RNA-seq library preparation

HUVEC total RNA was isolated using the Quick-RNA Microprep kit (Zymo Research) following the manufacturer's instructions. RNA's integrity and concentration were evaluated on 2100 Bioanalyzer (Agilent). Samples with a RIN >9 were selected for RNA-seq library preparation. Libraries were prepared from 1 $\mu$g of RNA using the TruSeq Stranded mRNA library prep kit (Illumina) following the reference guide. Synthesized libraries were sequenced on NextSeq 500 System (Illumina) with paired-end 75-bp reads. A summary of sequenced and mapped reads is in Table S1.

### ATAC-seq library preparation

Libraries for assay for transposase-accessible chromatin were prepared as described previously (Buenrostro et al, 2015). Briefly, 50,000 HUVECs were harvested, resuspended in ice-cold DPBS, and centrifuged (1 min at 1,300$g$, 4°C). Pelleted cells were resuspended in 50 $\mu$l of cold lysis buffer and centrifuged (10 min at 1,200$g$, 4°C). After discarding the supernatant, 50 $\mu$l of transposase mix with 2.5 $\mu$l of Tn5 transposase (Illumina) was added and incubated for 30 min at 37°C. The reaction was stopped by purification with the MinElute PCR purification kit (QIAGEN) following the manufacturer's instructions. Library amplification by PCR was done after determining the number of cycles needed. Products were size-selected with Agencourt AMPure XP beads (Beckman Coulter) and sequenced on NextSeq 500 System (75-bp paired-end reads; Illumina). Details are shown in Table S2.

### RNA-seq data analysis

Data quality was assessed using FastQC. Alignments of paired-end samples were generated using STAR v2.7.3a with default parameters using the human hg38 genome and annotations from the UCSC portal. Raw gene count matrices were generated using FeatureCounts with flags "-p -s 2." Differential gene expression was analyzed with DESeq2 v1.36; genes with an adjusted $P < 0.05$ and a $\log_2$ fold change > 0.5 and < −0.5 were considered differentially expressed. Pathway enrichment was done via the EnrichR web tool with a $P$-value cutoff of 0.05 to consider a term to be enriched. The Kyoto Encyclopedia of Genes and Genomes (KEGG) and the Wiki-Pathways databases were used for our analyses. Motif enrichment analysis in promoters of DEGs was done with the findMotifs tool from HOMER software with flags "-start −1,000 -end 100." Heatmaps were generated using DESeq2 variance-stabilizing transformation normalized counts and the R package Pheatmap. Volcano plots were generated using the R package EnhancedVolcano.

### ATAC-seq data analysis

Data quality was assessed with FastQC. Adapters were trimmed using TrimGalore, and alignment files were referenced to the human hg38 genome with Bowtie2 v2.3.5 with flag "−X 1,000" to align fragments up to 1 kb in length. Mitochondrial and PCR duplicate reads were discarded using Samtools and PicardTools, respectively. ENCODE blacklist-mapped reads were filtered via Bedtools. Peaks were called using MACS2 with flags "--broad --broad-cut-off 0.05 --keep-dup all." We generated a consensus peak set per treatment with the overlapping peaks from the two biological replicates of each treatment. Differentially accessible chromatin analysis was made using the getDifferentialPeaks tool from HOMER, where a peak region with a fold change >1.5 and a Poisson $P < 0.0001$ was considered to be differentially accessible. Signal plots and heatmaps were generated using the computeMatrix and plotProfile/plotHeatmap tools from Deeptools, peak annotation was made using the annotatePeaks tool from HOMER, and plots were generated using the log CPM normalized counts calculated with the edgeR package. Bigwig files for track visualization were generated using the bamCoverage tool from Deeptools. Public HUVEC H3K27me1 (ENCSR000AKL) and H3K27ac (ENCSR000ALB) bigwig and narrowPeak files were downloaded from ENCODE portal. The set of HUVEC putative enhancer regions was generated by filtering regions with overlapping H3K4me1, H3K27ac, and ATAC-seq peaks with the intersect tool from Bedtools. Transcription factor motif enrichment analysis in differentially accessible enhancers was made using the findMotifsGenome tool from the HOMER with a "-size 50" flag to limit to ±50 pb from the peak. Putative target genes for differential accessible enhancers were found using the HUVEC Hi-C contact matrix and topological domain coordinates obtained in a previous study by Rao et al (2014) to delimit the "regulatory domain" of each enhancer. Genomic track images were generated using pyGenomeTracks. Pathway enrichment analysis of putative target genes of DAEs was done using the KEGG database within the EnrichR web tool with a $P$-value cutoff of 0.05.

### NRF2 gene overexpression

The pHAGE-NFE2L2-IRES-EGFP plasmid (116765; Addgene) was used to produce lentiviral particles. The vector was also modified to have EGFP-only as a negative control. Briefly, $8 \times 10^6$ HEK293FT cells were seeded in a 15-cm dish with DMEM containing 10% FBS. The next

day, cells were transfected using Lipofectamine 3000 (Thermo Fisher Scientific) following the manufacturer's instructions: 20 µg of the transfer vector (with NFE2L2 or EGFP-only) and 10 µg of each packaging and envelope plasmids (psPAX2 and pMD2.G). The lentivirus-containing supernatant was collected at 48 and 72 h post-transfection and then concentrated using Amicon Ultra 100 kD filters (Merck). Lentiviral titer was calculated by the % of GFP+ cells using flow cytometry on serial dilutions of concentrated lentivirus stock. Transduction of HUVECs was done by spinoculation where the lentivirus supernatant was added to the medium (MOI = 20) with 10 µg/ml polybrene and the plate with cells was centrifuged at 800$g$ for 30 min at 32°C. Media were replaced after 24 h. This approach yielded a 90% transduction efficiency (measured by % GFP+ flow cytometry).

### RT–qPCR

HUVEC total RNA was isolated using Quick-RNA Microprep Kit (Zymo) following the manufacturer's instructions. cDNA was synthesized from 500 ng of RNA with qScript cDNA SuperMix (Quantabio). RT–qPCRs were run on CFX384 Real-Time Detection System (Bio-Rad). Reactions were prepared to a final 10 µl volume using 2X KAPA SYBR FAST RT-qPCR Master Mix, primers (200 nM each), 25 ng of cDNA, and nuclease-free water. The amplification program was as follows: 95°C for 3 min, followed by 40 cycles of 95°C for 3 s and 60°C for 30 s, followed by a melt curve. Expression levels were calculated as relative to the housekeeping gene $\beta$-actin. Primer sequences are shown in Table S3.

### Statistical analysis

Data are presented as the mean ± SD; statistical tests used in each case are indicated in figure legends; these tests were performed using either R or GraphPad Prism. One-way ANOVA was used to compare the means of our treatments, and pairwise comparisons were made with Tukey's test. The statistical significance cutoff used was a $P < 0.05$ unless stated otherwise.

## Data Availability

Raw and processed high-throughput sequencing data are available in the Gene Expression Omnibus repository under the accession number GSE241566.

## Supplementary Information

## Acknowledgements

M Wilson-Verdugo conducted this study to fulfill the requirements of Programa de Doctorado en Ciencias Bioquímicas of Universidad Nacional Autónoma de México (UNAM), and received a doctoral scholarship from Consejo Nacional de Humanidades, Ciencias y Tecnologías (#CVU 853375). We thank the UBM, UBMI, U de Computo, and Bioterio from the IFC-UNAM. We thank Professor Mayra Furlan-Magaril for critical reading of the article. This study was supported by PAPIIT-UNAM (IN203820), Premio de Investigación en Biomedicina Dr. Rubén Lisker 2018, and CONACYT CB (0284867) to VJ Valdes. M Wilson-Verdugo was supported by a CONACYT scholarship (754270).

## Author Contributions

M Wilson-Verdugo: conceptualization, data curation, formal analysis, investigation, methodology, and writing—original draft, review, and editing.
B Bustos-García: formal analysis, investigation, methodology, and writing—original draft.
O Adame-Guerrero: investigation and methodology.
J Hersch-González: investigation and methodology.
N Cano-Domínguez: formal analysis, investigation, methodology, and project administration.
M Soto-Nava: investigation and methodology.
CA Acosta: methodology.
T Tusie-Luna: resources, supervision, and funding acquisition.
S Avila-Rios: resources, supervision, and funding acquisition.
LG Noriega: resources and funding acquisition.
VJ Valdes: conceptualization, resources, data curation, formal analysis, supervision, funding acquisition, investigation, and writing—original draft, review, and editing.

## Conflict of Interest Statement

The authors declare that they have no conflict of interest.

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
