## [Reviewer comments · Life Science Alliance]

Life Science Alliance

Reversal of high-glucose-induced transcriptional and epigenetic memories through NRF2 activation

Marti Wilson-Verdugo, Brandon Bustos-Garcia, Olga Adame-Guerrero, Jaqueline Hersch-Gonzalez, Nallely Cano-Dominguez, Maribel Soto-Nava, Carlos Acosta, Teresa Tusie-Luna, Santiago Avila-Rios, Lilia Noriega, and Victor Valdes

DOI: <https://doi.org/10.26508/lsa.202302382>

Corresponding author(s): Victor Valdes, Universidad Nacional Autónoma de México

Review Timeline:

Submission Date:	2023-09-18
Editorial Decision:	2023-12-18
Revision Received:	2024-03-17
Editorial Decision:	2024-04-19
Revision Received:	2024-04-24
Editorial Decision:	2024-04-25
Revision Received:	2024-04-29
Accepted:	2024-04-30

Transaction Report:

December 18, 2023

Re: Life Science Alliance manuscript #LSA-2023-02382-T

Dr. Julian Valdes
Instituto de Fisiologia Celular
Cell Biology
Cto. Exterior s/n, C.U., Coyoacán
Mexico City 04510
Mexico

Dear Dr. Valdes,

Thank you for submitting your manuscript entitled "NRF2 pathway activation reverts high-glucose-induced transcriptional memory in endothelial cells" to Life Science Alliance. The manuscript was assessed by expert reviewers, whose comments are appended to this letter. We invite you to submit a revised manuscript addressing the Reviewer comments.

Thank you for this interesting contribution to Life Science Alliance. We are looking forward to receiving your revised manuscript.

Sincerely,

B. MANUSCRIPT ORGANIZATION AND FORMATTING:

Reviewer #1 (Comments to the Authors (Required)):

I have had the opportunity to conduct a thorough review of the manuscript submitted by Wilson-Verdugo et al., which investigates the metabolic and transcriptional impacts of transient elevated glucose exposure on human endothelial cells. The study provides crucial insights into the persistent transcriptional and chromatin accessibility changes induced by transient high glucose exposure in endothelial cells, highlighting the role of the NRF2 pathway in potentially reversing these effects.

This manuscript is indeed a valuable contribution to the existing body of knowledge, considering the clinical relevance and comprehensiveness of the study. However, there are several aspects that require attention and clarification before this work can be considered for publication. My detailed comments and recommendations are outlined below.

Major Concerns:

Control Experiments:

- The study would benefit significantly from the inclusion of an osmotic control group consisting of 5.5 mM glucose and 25.5 mM mannitol. This would serve to confirm that the observed differences are not a consequence of variations in osmotic pressure. Additionally, it is imperative to incorporate this osmotic control in the second phase of the treatment with low glucose for the transient high glucose group. While replicating the ATAC-seq and RNA-seq data for these control groups may be impractical, at the very least, the metabolic changes should be demonstrated.

Statistical Analysis:

- The utilization of multiple unpaired t-tests in figures 1, 4, and supplemental figure S4 to compare between groups is not advisable due to the increased risk of Type 1 errors. A more suitable approach would be to employ a One-way ANOVA for these comparisons.

Functional Assays:

- The manuscript would be substantially strengthened by the inclusion of additional functional assays. I recommend incorporating measurements of NO production, mitochondrial ROS (e.g. MitoSOX: Marks et al. Sci Rep. 2015 Jul 14;5:11427; Mone et al. Cells 2021;10(8):2115), global cellular ROS, and oxidative DNA damage (8-OxoG, e.g. Nakabeppu Y et al. Sci Rep 2016;6:22086). Assessing the impact of NRF2 overexpression and sulforaphane supplementation on these assays would also be beneficial.

Minor Concerns:

- In line 75, the assertion that GLUT-1 expression remains unchanged under high glucose conditions is somewhat oversimplified and supported by a dated reference. While mRNA expression may remain constant, protein expression on the cell membrane is reduced in HUVECs (Tumova et al. 2016: <https://doi.org/10.1016/j.vph.2016.11.001>). Additionally, primary vascular endothelial cells from other mammals exhibit downregulation of GLUT-1 under high glucose conditions (Totary-Jain et al. 2005: <https://doi.org/10.1161/01.RES.0000189260.46084.e5> and Alpert et al. 2005: <https://doi.org/10.1007/s00125-005-1681-y>). I would recommend some more nuance in the phrasing.

- The Venn Diagrams in Figure 2c and Figure 3a would be more effective in communicating the data if they were constructed to be area-proportional.

- The relevance of the results pertaining to Vascular Smooth Muscle Contraction and Diabetic Cardiomyopathy in Figure 3f should be scrutinized, given that the analysis was conducted in endothelial cells.

- The access number provided for the Gene Expression Omnibus repository is currently set to private, with a release date of August 1, 2026. Ensuring that the data is made available at the time of publication is crucial.

I believe that addressing these concerns and recommendations will significantly enhance the quality and robustness of the manuscript, making it a worthy addition to the scientific community. I look forward to seeing the revised version of this

manuscript and am confident that the authors will be able to address the highlighted issues satisfactorily.

Reviewer #2 (Comments to the Authors (Required)):

Summary

In this paper, the authors culture endothelial cells in low and high glucose conditions, with some of the high glucose cells reverted back to low glucose culture to test metabolic memory. They then performed metabolic and oxidative stress assays, as well as transcriptomics and chromatin accessibility analysis. The results show interesting transcriptomic and chromatin accessibility changes in both the high glucose and the memory condition, and either overexpression or chemical stimulation of NRF2 partially abrogates these effects. Overall, the paper is well written, and the evidence on transcriptomic and accessibility changes with glycemic memory, as well as the role of NRF2, are advances in the field. However, the data are limited due to the low number of samples, the sole focus on one set of pooled HUVEC, and the extreme glucose levels selected. The authors should also determine if these transcriptomic changes induced by NRF2 then result in functional changes of the cells.

Comments

1. The introduction is well-written and clear.
2. Could the authors provide more details on the pooled-donor HUVEC? How many donors, and are there any data on donor sex?
3. The data should be replicated in another human endothelial cell type (e.g., coronary artery) so that we know these effects are not specific to HUVEC.
4. Why were only 5 and 30 mM glucose used for these time points? 30 mM is extremely high. Could you do assays to determine the effect of a more reasonable level of glucose (e.g., 15 mM)? And also assays to determine at what time point these effects occur.
5. Please clarify what is meant by biological vs. technical replicates in terms of cell culture samples arises in Figure 1b and d, as well as in Supplemental Figure 1. For all of the transcriptomics, what is meant by biological replicates? Are the biological replicates from the same cell pool but in different plates or wells? Were measurements on the same day or different days?
6. What is the justification for using an unpaired t-test for analysis? T-tests are reserved for samples that are normally distributed, and with a sample size of 3 in many of the assays, it would not be possible to run a test to establish normality. A non-parametric statistical test is preferred for data that cannot be shown to be normally distributed, and that would certainly require more samples (6-9).
7. Could the authors explain why they chose to use Dihydrorhodamine123 for the ROS assay?
8. Was the change in cell cycle statistically significant (Figure 1c)?
9. The text in some of the figures is difficult to read. Please make the text bigger.
10. Could the authors include normal glucose samples with NRF2 overexpression and sulforaphane treatment as well as controls?
11. The meaning of Figure 4e is unclear. Please explain this better in the text and caption.
12. Why does the analysis of NRF2 overexpression and sulforaphane treatment stop with expression analysis? The authors should measure functional outcomes of hyperglycemia and memory, for example the ROS shown in Figure 1d, to see if these treatments help with the functional changes.
13. Some of the discussion is repetitive of the results. Try to avoid repeating results and instead focus on discussing what the results mean in the context of the literature.

We want to express our sincere gratitude to the reviewers for their insightful and constructive feedback on our manuscript, which undoubtedly enhanced the quality of our work. Below, we will address each point raised:

Reviewer #1 (Comments to the Authors (Required)):

Major Concerns:

Control Experiments:

- The study would benefit significantly from the inclusion of an osmotic control group consisting of 5.5 mM glucose and 25.5 mM mannitol. This would serve to confirm that the observed differences are not a consequence of variations in osmotic pressure. Additionally, it is imperative to incorporate this osmotic control in the second phase of the treatment with low glucose for the transient high glucose group. While replicating the ATAC-seq and RNA-seq data for these control groups may be impractical, at the very least, the metabolic changes should be demonstrated.

RESPONSE = We thank the reviewer for his or her suggestion. We have now incorporated osmotic control samples into the Seahorse (metabolic) and RT-qPCR experiments, which are now presented in Supplemental Figures S1a and S4a, respectively. These treatments comprised: culture of HUVEC with 5.5 mM glucose + 25 mM mannitol, named “control + mannitol”; “memory + mannitol”, where we added 5.5 mM glucose + 25 mM mannitol in the second phase of the treatment; and “memory + mannitol + SF”, where we added 5.5 mM glucose + 25 mM mannitol + 1 μM sulforaphane in the second phase of the treatment. Our analyses indicated that the addition of mannitol as an osmotic control did not alter the expression of the reported genes or the metabolic profile of the cells under the experimental conditions.

Given the absence of differences provoked by mannitol addition in these experiments, and considering previous studies of the in vitro metabolic memory where an osmotic control was used, we believe it is safe to assume the memory effects we observe are not affected or dependent on increased osmolarity. We believe it would be beneficial to further explore the role of osmolarity per se in future studies, but for now such exploration is beyond the scope of our current study.

Something worth mentioning, is that thanks to Reviewers insightful suggestions regarding metabolic measurements with the osmotic control, we refined our Seahorse protocol by normalizing the data based on cell numbers using nuclei quantification with a BioTek Cytation 1 system, replacing the original method of protein concentration determination by Bradford assay. Following this protocol enhancement and conducting nine biological replicates on nine different days with multiple technical replicates each, we did not observe metabolic differences in our treatments opposite to what other authors have reported regarding oxygen consumption (PMID: 33483741). These results are detailed in Supplementary Figure 1a. Consequently, we have revised our terminology in the results section, focusing solely on epigenetic and transcriptional memory while excluding indications to metabolic memory. Importantly, this adjustment does not impact our conclusions regarding the transcriptional and epigenetic changes induced by transient high glucose in endothelial cells or the effectiveness of NRF2 pathway activation in reversing them. We appreciate the reviewer's guidance that allowed us to make these corrections.

Statistical Analysis:

- The utilization of multiple unpaired t-tests in figures 1, 4, and supplemental figure S4 to compare between groups is not advisable due to the increased risk of Type 1 errors. A more suitable approach would be to employ a One-way ANOVA for these comparisons.

RESPONSE = We appreciate the reviewer's valuable input regarding statistical analysis. Accordingly, to reviewer's suggestion, we have transitioned our statistical analyses from t-tests to One-way ANOVA for group comparisons, and for pairwise comparisons we used the Tukey's post-hoc test when the results from the ANOVA were statistically significant. Worth mentioning that statistical significance was still reached in all cases after this new analysis, thus, previous interpretations and conclusions were not modified from the previous version

Functional Assays:

- The manuscript would be substantially strengthened by the inclusion of additional functional assays. I recommend incorporating measurements of NO production, mitochondrial ROS (e.g. MitoSOX: Marks et al. Sci Rep. 2015 Jul 14;5:11427; Mone et al. Cells 2021;10(8):2115), global cellular ROS, and oxidative DNA damage (8-OxoG, e.g. Nakabeppu Y et al. Sci Rep 2016;6:22086). Assessing the impact of NRF2 overexpression and sulforaphane supplementation on these assays would also be beneficial.

RESPONSE = We sincerely appreciate the valuable feedback from reviewer #1, as this suggestion significantly enhances the manuscript. In response to his/her recommendation, we incorporated MitoSOX and dihydrorhodamine 123 (DHR 123) for ROS measurement, along with DAF-2 DA for NO measurement; these results are now presented in Figure 1d. We found that, while there was no significant difference in NO concentrations, both high-glucose and the memory conditions led to an elevation in MitoSOX and DHR 123 signals. Regrettably, due to a delay in the availability (backorder) of the 8-OxoG reagent, we were unable to assess oxidative DNA damage as planned.

As advised by reviewer, we also evaluated the effect of SF supplementation in ROS production measured by MitoSOX and DHR 123; the results are shown in Figure 5e. We found that, in the case of DHR 123 signal, SF supplementation decreased ROS levels in both high-glucose and memory. Meanwhile, in the case of the MitoSOX assays, we found that SF supplementation decreased ROS signal in the high-glucose condition, but not in the memory condition; we hypothesize that this may be due to differences in the clearance of different ROS after SF supplementation, since DHR 123 predominantly reacts with peroxynitrite, while MitoSOX reacts predominantly with superoxide. Nonetheless, these experiments show that HUVEC have a persistent increase in ROS after an episode of transient high glucose that can be reduced to some extent with SF supplementation. Even though SF supplementation did not completely revert the oxidative memory, these new results represent important information that improves our work. Note: in the case NO assays, we did not test SF supplementation since we did not observe changes induced by our high-glucose treatments (Figure 1d).

Moreover, to address the importance of evaluating the effect of SF supplementation in the persistent chromatin accessibility changes that we found in response to transient high glucose exposure, we performed and included data analyses of ATAC-seq of the memory + SF condition

in the manuscript. These new analyses are presented in Figure 5 panels a-d, demonstrating that 80% of the memory differential accessible regions were restored with SF supplementation. This information shows that SF supplementation not only restored the transcriptional changes but also mitigated most of the epigenetic alterations associated with HG-induced memory. Based on this significant information, we have updated the article title to better reflect these latest findings.

Minor Concerns:

- In line 75, the assertion that GLUT-1 expression remains unchanged under high glucose conditions is somewhat oversimplified and supported by a dated reference. While mRNA expression may remain constant, protein expression on the cell membrane is reduced in HUVECs (Tumova et al. 2016: <https://doi.org/10.1016/j.vph>). Additionally, primary vascular endothelial cells from other mammals exhibit downregulation of GLUT-1 under high glucose conditions (Totary-Jain et al. 2005: <https://doi.org/10.1161/01>. and Alpert et al. 2005: <https://doi.org/10.1007/>). I would recommend some more nuance in the phrasing.

RESPONSE = We thank the reviewer for his/her time to provide these references. We concur with the reviewers' comment. We have corrected the phrasing of this sentence to "The vascular system is particularly sensitive to blood glucose concentrations, as endothelial cells (ECs) are in direct contact with the bloodstream and display increased susceptibility due to the predominant expression of insulin-independent glucose transporter GLUT-1" (now in line 54)

- The Venn Diagrams in Figure 2c and Figure 3a would be more effective in communicating the data if they were constructed to be area-proportional.

RESPONSE = We thank the suggestion and we have replaced the previous Venn diagrams to area-proportional Venn diagrams in the mentioned Figure panels to better communicate the data.

- The relevance of the results pertaining to Vascular Smooth Muscle Contraction and Diabetic Cardiomyopathy in Figure 3f should be scrutinized, given that the analysis was conducted in endothelial cells.

RESPONSE = We thank the reviewer for noticing this. The terms mentioned on Figure 3f were "gene ontology terms" for which it is not uncommon to retrieve terms that are not directly related to the cells or the specific processes studied, as GO terms are based on comparison to previously curated lists of genes (PMID: 10802651). However, we re-analyzed the data for this panel, this time using the HG+M DAEs subset instead of just the HG/M-shared DAEs to improve the narrative of the manuscript, this to serve as introduction for the new analyses performed in Figure 5 panels a-d, where we analyzed ATAC-seq data of HUVEC exposed to the memory + SF condition.

- The access number provided for the Gene Expression Omnibus repository is currently set to private, with a release date of August 1, 2026. Ensuring that the data is made available at the time of publication is crucial.

RESPONSE = Certainly, we have deposited all the high-throughput sequencing data, including the new ATAC-seq samples with SF supplementation, in the Gene Expression Omnibus, and we will make it available for download immediately upon manuscript acceptance. Of course, we are

more than willing to provide the data to the reviewers should they express interest in examining it prior to publication.

Reviewer #2 (Comments to the Authors (Required)):

Comments

1. The introduction is well-written and clear.

We thank reviewer #2 for his/her positive words.

2. Could the authors provide more details on the pooled-donor HUVEC? How many donors, and are there any data on donor sex?

RESPONSE = Certainly. We thank the reviewer for the opportunity to clarify this important point. To minimize variability between donors we performed all the experiments with pooled HUVEC cells from the company Lonza. According to the information provided by Lonza, pooled cell vials come from 3-6 donors (<https://knowledge.lonza.com/faq?id=756&search=donor+huvec>). We looked for the donor information of the vials we used in this study, and they come from mixed sex. The lot numbers (0000478982, 18TL169251 and 23T2043669) are now specified in the method section of the manuscript (line 507).

3. The data should be replicated in another human endothelial cell type (e.g., coronary artery) so that we know these effects are not specific to HUVEC.

RESPONSE = While we acknowledge the importance of replicating our experiments in other cell types to validate the metabolic memory across different lineages, the metabolic memory phenomenon has been well-documented in multiple cell types, including different types of endothelial cells, such as aortic (doi: 10.1161/CIRCRESAHA.112.266593) and retinal (doi: 10.1186/s12886-018-0921-0). It has also been reported in immune cells (doi: 10.4049/jimmunol.1901348; and doi: 10.1161/CIRCULATIONAHA.120.046464), vascular smooth muscle cells (doi: 10.1073/pnas.0803623105), fibroblasts (doi: 10.4161/15592294.2014.967584), and glomerular mesangial cells (doi:10.1159/000493619), among others. Considering this body of evidence, and our lack of access to another human endothelial cell type in Mexico, we decided to expand our study to *in vivo* aorta samples. To this end, we attempted to induce a metabolic memory phenotype in mice. We had two groups: control diet group (n=6; 3 males and 3 females) and the high fat diet group (HFD; n=8; 4 males and 4 females). We fed the mice ad-libitum with the corresponding diet for 8 weeks (Letter Figure 1), the time at which we performed a glucose tolerance test, this by first fasting the mice for 6 hours, then we injected 2 g/kg glucose intraperitoneally and made repeated measurements of blood glucose through the course of 120 min (Letter Figure 2). We found that after 8 weeks, the HFD group had developed a marked glucose intolerance compared with the control group (Letter Figure 1C). Then we proceeded to separate the control group into two subgroups: the control + vehicle group (n=3), where the mice were injected subcutaneously with PBS every 48 hours for 4 weeks; and the control + SF group (n=3), with subcutaneous injections of 5 mg/kg sulforaphane. We also separated the HFD into two subgroups: the memory + vehicle group (n=4), where we replaced the HFD with control diet, and subcutaneous injection of PBS was performed every 48 hours for 4 weeks; and the memory + SF group (n=4), where we replaced the HFD with the control diet, and we injected subcutaneously 5 mg/kg sulforaphane every 48 h for 4 weeks (Letter Figure 1A).

Letter Figure 1. A) Experimental design to induce a diet-induced metabolic memory phenomenon in mice. B) Weight curves of mice fed with control or high fat diet at 8 weeks. C) Glucose tolerance test at 8 weeks.

After the 16 weeks of the experiment, we euthanized the mice and dissected their aortas. Subsequently, we isolated RNA and prepared RNA-seq libraries of each individual aorta. After sequencing, we assessed the PCA distribution of the samples and performed differential gene expression analyses between groups. Unfortunately, we did not detect any transcriptional changes between our memory treatments and the control (Letter Figure 3) revealing the failure of our protocol. It is evident that our experimental design must be improved to induce metabolic memory in mice, probably by exposing the animals to longer periods of HFD (up to 6 months as other authors have done) aided with supplementation of the HFD with sucrose in the water. Additionally, we will need to adjust concentration and bioavailability of SF *in vivo*. Overall, we hope the Reviewer #2 recognize our efforts to replicate the metabolic memory phenomenon *in vivo* and our willingness to prove the effect of SF to ameliorate the metabolic memory in mice aortas, but for now, repeating the experiments *in vivo* is not feasible for us due to limitations in time and resources.

Letter Figure 3. Principal component analysis and volcano plots of RNA-seq data from

aorta biopsies. Animals were subjected to a metabolic memory protocol with or without Sulforaphane (SF) supplementation for 8 weeks. All comparisons were performed versus control animals.

4. Why were only 5 and 30 mM glucose used for these time points? 30 mM is extremely high. Could you do assays to determine the effect of a more reasonable level of glucose (e.g., 15 mM)? And also assays to determine at what time point these effects occur.

RESPONSE = This is a valid question from reviewer #2. We concur that 30 mM glucose is on the upper end of what can be found in patients with diabetes. We included two references in the manuscript where glucose levels above 20 mM are reported in patients with uncontrolled diabetes (doi: <https://doi.org/10.2337/diacare.20.9.1353> and doi: [10.1530/acta.0.1180365](https://doi.org/10.1530/acta.0.1180365)). As the reviewer asked, we included qRT-PCR experiments with 15 mM glucose in Figure 1b, observing that 4 days in 15 mM glucose had no effect on the 6 genes examined, and that 8 days in 15 mM glucose had an effect on only one gene. Thus, while we cannot rule out the possibility that longer times could induce an effect, it appears that 15 mM glucose is insufficient to induce an effect similar to what we observe with 30 mM after 4 days *in vitro*, making the 15 mM condition impractical due to the fact that primary cell lines (such as HUVEC) can be cultured for a limited number of cell divisions.

Additionally, to our knowledge, the timeframe when these changes take place *in vitro* is likely very short. It has been reported that as short as 16 hours in high glucose can induce persistent transcriptional and epigenetic alterations in endothelial cells *in vitro* (doi: [10.1084/jem.20081188](https://doi.org/10.1084/jem.20081188)).

We agree with the Reviewer that would be relevant to determine the time point when these changes induce a “memory” effect, and also to determine how long does that memory last. In that regard, and as Reviewer requested, we examined the duration of the “memory” effect after 4 days in 30 mM glucose via RT-qPCR, finding that after 8 days in normal glucose, half of the assessed genes returned to control levels, suggesting a wavering of the transcriptional memory. The results are included in Figure 1b.

5. Please clarify what is meant by biological vs. technical replicates in terms of cell culture samples arises in Figure 1b and d, as well as in Supplemental Figure 1. For all of the transcriptomics, what is meant by biological replicates? Are the biological replicates from the same cell pool but in different plates or wells? Were measurements on the same day or different days?

RESPONSE = We are happy to clarify: biological replicates are cells that come from independent wells/plates cultured in the same glucose or SF treatment, and technical replicates come from measuring the same sample multiple times. E.g, in the case of RT-qPCR we used three biological replicates, with three technical replicates each; and for the Seahorse assays we used 9 biological replicates, with 6 technical replicates each. For flow cytometry (cell viability, ROS, NO) assays, we performed biological replicates in different days, each one with different wells and averaged the fluorescence intensity of 5000-10000 individual cells in each sample. Similarly, for the Seahorse experiments we used 9 biological replicates coming from experiments performed in 9 different days, and the same well was subcultured in 6 different wells in the Seahorse plate the day before the assay to serve as the 6 technical replicates. Meanwhile, in the high-throughput sequencing experiments, we cultured and prepared the replicates on the same days to reduce the risk of having batch effects, but each sample was independently processed throughout the process, from cell culture to library preparation and sequencing. A better explanation of biological vs technical replicates is now included in the method section of the manuscript.

6. What is the justification for using an unpaired t-test for analysis? T-tests are reserved for samples that are normally distributed, and with a sample size of 3 in many of the assays, it would not be possible to run a test to establish normality. A non-parametric statistical test is preferred for data that cannot be shown to be normally distributed, and that would certainly require more samples (6-9).

RESPONSE = We thank Reviewer #2 for his/her observation. We changed the statistical analyses used for our data to one-way ANOVA as suggested by Reviewer #1.

7. Could the authors explain why they chose to use Dihydrorhodamine123 for the ROS assay?

RESPONSE = We used dihydrorhodamine 123 for our assays since it reacts predominantly with peroxynitrite, and this species has been highlighted as one important contributor to the oxidative damage and vascular dysfunction in diabetes (doi: 10.1155/2007/21976, 10.2174/092986711794088317 and doi: 10.2337/db13-0577 for reference). Nonetheless, we also included ROS assays with MitoSOX, which predominantly reacts with superoxide, in Figures 1d and 5e.

8. Was the change in cell cycle statistically significant (Figure 1c)?

RESPONSE = Yes, the change is statistically significant. We changed the type of graph to better communicate the data. We thank the reviewer for the comment that allow us to correct this in the figure.

9. The text in some of the figures is difficult to read. Please make the text bigger.

RESPONSE = We made a general adjustment of text size in the Figures. We hope that now the text is easier to read.

10. Could the authors include normal glucose samples with NRF2 overexpression and sulforaphane treatment as well as controls?

RESPONSE = We understand the reviewer's suggestion for including normal glucose + NRF2-OE/SF samples as controls as this could have potential unexpected positive or negative effects. However, our primary objective and efforts were focused to evaluate the effect of NRF2 activation in the context of our pathological high-glucose/memory treatments, comparing them to a "healthy" control without any supplementation. This approach is intended to better simulate the scenario where healthy individuals are unlikely to take any form of medication or supplementation prior to the onset of diabetes. While we acknowledge the relevance of assessing the (beneficial or detrimental) effects of NRF2 activation in a normoglycemic context, our current resources and economic limitations do not allow us to explore these avenues at this time. However, we agree that it would be a valuable area to investigate in future studies.

11. The meaning of Figure 4e is unclear. Please explain this better in the text and caption.

RESPONSE = We appreciate Reviewer's feedback. We have removed the mentioned panel for clarity of the manuscript.

12. Why does the analysis of NRF2 overexpression and sulforaphane treatment stop with expression analysis? The authors should measure functional outcomes of hyperglycemia and memory, for example the ROS shown in Figure 1d, to see if these treatments help with the functional changes.

RESPONSE = This is a very important point raised by Reviewer #2, and we appreciate him/her for its insightful feedback. In response, we have expanded our analysis beyond gene expression to include functional outcome measurements related to hyperglycemia and memory. We have incorporated additional ROS measurement assays using MitoSOX and dihydrorhodamine 123 (DHR123), as well as NO assays with DAF-2 DA, now presented in Figure 1d. Our findings indicate that while there was no significant difference in NO production, both high-glucose and the memory conditions led to an elevation in MitoSOX and DHR 123 signals. Subsequently, in line with Reviewer #2 suggestions, we conducted DHR123 and MitoSOX assays in HUVEC supplemented with SF. We found that, at least in the case of DHR 123 signal, SF supplementation decreased ROS levels in both high-glucose and memory. Meanwhile, in the case of the MitoSOX assays we found that SF supplementation decreased ROS signal in the high-glucose condition, but not in the memory condition; we hypothesize that this may be due to differences in the clearance of different ROS by SF, since DHR 123 predominantly reacts with peroxynitrite, while MitoSOX reacts predominantly with superoxide. Nonetheless, these experiments show that HUVEC have a persistent increase in ROS after an episode of transient high glucose that can be mitigated to some extent with SF supplementation. These results are shown in Figure 5e. Although SF supplementation did not completely revert the oxidative memory, this new finding represents important information that enhanced our work and thus we thank the Reviewer for the valuable suggestion.

In addition to what we have mentioned in the response to your kind suggestions to improve this study, and to address the importance of evaluating the impact of SF supplementation on the persistent chromatin accessibility changes that we found in response to transient high glucose exposure, we performed and included analyses of ATAC-seq data of the memory + SF condition in the manuscript. You will find these new analyses in Figure 5 panels a-d, where we show that 80% of the memory differentially accessible regions were restored by SF supplementation. We consider that these results represent an important piece of information, which highlights that SF supplementation not only reversed transcriptional changes but also most of the epigenetic alterations that persisted after an episode of transient high glucose. Based on this significant data, we have updated the title of the article to better reflect these latest findings.

13. Some of the discussion is repetitive of the results. Try to avoid repeating results and instead focus on discussing what the results mean in the context of the literature.

RESPONSE = Once more, we thank the reviewer for his or her feedback aimed at enhancing the clarity of our manuscript. In response, we have reorganized the discussion section to enhance conciseness and have eliminated redundant discussions that overlap with the results section.

April 19, 2024

Re: Life Science Alliance manuscript #LSA-2023-02382-TR

Dr. Victor Julian Valdes
Universidad Nacional Autónoma de México
Instituto de Fisiología Celular. Cell Biology Department
Cto. Exterior s/n, C.U., Coyoacán
Mexico City, Mexico City 04510
Mexico

Dear Dr. Valdes,

Thank you for submitting your revised manuscript entitled "Reversal of high-glucose-induced transcriptional and epigenetic memories through NRF2 activation" to Life Science Alliance. The manuscript has been seen by the original reviewers whose comments are appended below. While the reviewers continue to be overall positive about the work in terms of its suitability for Life Science Alliance, some important issues remain.

Our general policy is that papers are considered through only one revision cycle; however, we are open to one additional short round of revision. Please note that I will expect to make a final decision without additional reviewer input upon re-submission.

Please submit the final revision within one month, along with a letter that includes a point by point response to the remaining reviewer comments.

To upload the revised version of your manuscript, please log in to your account: <https://lsa.msubmit.net/cgi-bin/main.plex>
You will be guided to complete the submission of your revised manuscript and to fill in all necessary information.

B. MANUSCRIPT ORGANIZATION AND FORMATTING:

Sincerely,

Reviewer #1 (Comments to the Authors (Required)):

The authors have successfully addressed all my comments, and the current version of the manuscript has been significantly

improved. I would particularly like to commend the authors for the refinement of the Seahorse protocol, which led to the disappearance of the metabolic effects reported in the previous version of the manuscript. It is always challenging to see an effect disappear after the refinement of a method, and reporting this honestly should be positively recognized.

However, I have identified a potentially serious issue with the interpretation of some of the added results in Supplemental Figure S4a. If I understand correctly, the Memory + Mannitol group depicted in the figure underwent 4h of 5mM Glucose + 25mM Mannitol followed by 4h of 5mM Glucose, whereas the Control + Mannitol group was subjected to 8h of 5mM Glucose + 25mM Mannitol. This approach was taken following my suggestion to investigate the effects of higher osmolar pressure in comparison to the presence of glucose. In the presented results, the Control + Mannitol group showed no differences in gene expression compared to the Control group, suggesting that the changes observed in the High Glucose group are indeed attributable to the presence of glucose. Conversely, the gene expression in the Memory + Mannitol group differs significantly from the Control group and closely resembles that of the Memory group, suggesting that the effects on gene expression observed in the Memory group are due to changes in osmotic pressure rather than the transient presence of glucose. I would appreciate it if the authors could clarify whether my interpretation of the experiments is incorrect.

In their responses to my queries, the authors state, "Given the absence of differences provoked by the addition of mannitol in these experiments, and considering previous studies of in vitro metabolic memory where an osmotic control was used, we believe it is safe to assume the memory effects we observe are not influenced or dependent on increased osmolarity." If my interpretation is accurate, I must disagree; these results seriously challenge the authors' interpretation, and the changes in osmolarity might explain the memory effect observed. The authors should include a mannitol control in all the experiments they present and revise the manuscript accordingly.

Reviewer #2 (Comments to the Authors (Required)):

In this paper, the authors culture endothelial cells in low and high glucose conditions, with some of the high glucose cells reverted back to low glucose culture to test metabolic memory. They then performed metabolic and oxidative stress assays, as well as transcriptomics and chromatin accessibility analysis. The results show interesting transcriptomic and chromatin accessibility changes in both the high glucose and the memory condition, and either overexpression or chemical stimulation of NRF2 partially abrogates these effects. In revision, the authors included an osmotic control and tested a medium, more physiologically relevant glucose level; added more functional assays, including NO and ROS; updated their statistics; and added functional assays after NRF2 stimulation. The paper is improved in the revision, and the authors were responsive to the review comments. There remain a few concerns.

1. Many of the assays still only have 3 samples (biological replicates) in the same set of pooled HUVEC. All samples were measured at the same time. The paper would be stronger, and the results would be more robust, if a higher number of samples were presented (6-9 biological replicates) and these were measured in at least two experiments.
2. There are some aspects of the paper that the authors were not able to improve, and these should be added to the limitations paragraph. These include: using only HUVEC and not a more physiologically relevant endothelial cell type; seeing little effect at 15 mM glucose; not analyzing NRF2 activation in normal glucose samples; and the limitations of some of the functional measurements, like DAF and the ROS assays. In addition, all these assays were done on endothelial cells in vitro in static culture, which may have limited relevance to endothelial cells in vivo that are exposed to shear stress from blood flow.

SECOND RESPONSE TO REVIEWERS

Wilson-Verdugo et al.

Reviewer #1

The authors have successfully addressed all my comments, and the current version of the manuscript has been significantly improved. I would particularly like to commend the authors for the refinement of the Seahorse protocol, which led to the disappearance of the metabolic effects reported in the previous version of the manuscript. It is always challenging to see an effect disappear after the refinement of a method, and reporting this honestly should be positively recognized.

RESPONSE: Once more we appreciate the dedication and time of the Reviewer 1.

I have identified a potentially serious issue with the interpretation of some of the added results in Supplemental Figure S4a. If I understand correctly, the Memory + Mannitol group depicted in the figure underwent 4h of 5mM Glucose + 25mM Mannitol followed by 4h of 5mM Glucose, whereas the Control + Mannitol group was subjected to 8h of 5mM Glucose + 25mM Mannitol. This approach was taken following my suggestion to investigate the effects of higher osmolar pressure in comparison to the presence of glucose. In the presented results, the Control + Mannitol group showed no differences in gene expression compared to the Control group, suggesting that the changes observed in the High Glucose group are indeed attributable to the presence of glucose. Conversely, the gene expression in the Memory + Mannitol group differs significantly from the Control group and closely resembles that of the Memory group, suggesting that the effects on gene expression observed in the Memory group are due to changes in osmotic pressure rather than the transient presence of glucose. I would appreciate it if the authors could clarify whether my interpretation of the experiments is incorrect.

RESPONSE: We would like to clarify the concern raised by Reviewer #1. In his comment, the Reviewer initially states that "*If I understand correctly, the Memory + Mannitol group depicted in the figure underwent 4h of 5mM Glucose + 25mM Mannitol followed by 4h of 5mM Glucose*", and then gives his interpretation of the experiments depicted in Supplemental Figure S4a based on this claim. We apologize, **but this is an incorrect understanding of the culture conditions that we used in the Memory + Mannitol treatment:**

This condition was performed as specifically requested by the Reviewer 1 ("*...incorporate this osmotic control in the second phase of the treatment with low glucose for the transient high glucose group.* "). Thus, "Memory + Mannitol" comprised the culture of HUVEC for 4 days in 30 mM glucose (not 5.5 mM glucose + 25 mM mannitol as stated by reviewer #1) followed by 4 days in 5.5 mM glucose + 25 mM mannitol (**not 5.5 mM glucose as stated by reviewer #1**). Thus, this treatment, together with the Control + Mannitol (8 days in 5.5 mM glucose + 25 mM mannitol), allowed us to evaluate the effect of osmolarity in gene expression by comparing the results with those of the Control and Memory

treatments, which are identical to their counterparts except for the addition of mannitol (Letter Figure 1).

Letter Figure 1. Comparison of HUVEC culture conditions for Control, Control + Mannitol, Memory and Memory + Mannitol treatments. The Control group comprised the culture of HUVEC for 8 days in 5.5 mM glucose, while the Control + Mannitol group comprised the culture of HUVEC for 8 days in 5.5 mM glucose + 25 mM mannitol. On the other hand, the Memory group comprised the culture of HUVEC in 30 mM glucose for 4 days, followed by 4 days of culture in 5.5 mM glucose; while the **Memory + Mannitol group comprised the culture of HUVEC in 30 mM glucose for 4 days, followed by 4 days in 5.5 mM glucose + 25 mM mannitol.** These treatments allowed us to assess the effect of osmolarity in our gene expression experiments.

Having clarified this, we hope that it becomes clear that our results depicted in the Supplemental Figure S4a show:

- 1) There were no expression differences observed between the Control group and Control + Mannitol group in any of the genes evaluated. Therefore, we concluded that osmolarity (mannitol) has no effect on gene expression in HUVEC after 8 days of culture.
- 2) There were no expression differences observed between the Memory group and the Memory + Mannitol group; conversely, both Memory and Memory + Mannitol were different than the Control and Control + Mannitol groups. Hence, we concluded that the transcriptional memory that we observe in HUVEC is induced by high glucose exposure, and this effect is independent of the addition of mannitol.

To avoid further confusion, we added a more specific description of each of the treatments used for this panel in the legend of the Supplemental Figure S4a. Now it says: "Memory + Mannitol (4 days in 30 mM glucose followed by 4 days in 5.5 mM glucose + 25 mM mannitol)"

In their responses to my queries, the authors state, "Given the absence of differences provoked by the addition of mannitol in these experiments, and considering previous studies of in vitro metabolic memory where an osmotic control was used, we believe it is safe to assume the memory effects we observe are not influenced or dependent on increased osmolarity." If my interpretation is accurate, I must disagree; these results seriously challenge the authors' interpretation, and the changes in osmolarity might explain the memory effect observed. The authors should include a mannitol control in all the experiments they present and revise the manuscript accordingly.

RESPONSE: We hope that the explanation given above is enough to clarify the issue raised by the Reviewer. Since it was a matter of misunderstanding of the culture conditions used for the Memory + Mannitol group, we believe that the inclusion of more mannitol controls would not improve or substantially change any of the results and conclusions of this study.

Reviewer #2

We appreciate the Reviewer #2 recognition of our multiple efforts to improve or work.

- 1. Many of the assays still only have 3 samples (biological replicates) in the same set of pooled HUVEC. All samples were measured at the same time. The paper would be stronger, and the results would be more robust, if a higher number of samples were presented (6-9 biological replicates) and these were measured in at least two experiments.**

RESPONSE. We agree with the reviewer that having more biological replicates is always positive to strengthen the study. Though in our study, we have integrated experiments from different sets (days) or batches of cells (four) spanning over a four years period. For instance, RT-qPCR experiments of various genes were initially conducted in 2019-2020, followed by a repeat analysis done in the RNA samples used for RNA-seq (2022), and subsequently repeated during the revision process (2024). Similar approaches were undertaken for ROS and functional assays. Regarding NGS data, having more than duplicates or triplicates for all our different treatments is financially impossible for us. Nevertheless, our NGS samples were generated in different batches/years (i.e. initial assessment of high glucose effect libraries, NRF2 overexpression libraries, and the sulforaphane supplementation libraries were all made with different batches of cells and were made at least 6 months apart of each other). Unfortunately, at this point, it would be impossible for us to add more replicates given the current time constraints. But

more importantly, we believe that adding more replicates would not substantially alter any results or interpretations presented in the manuscript.

2. There are some aspects of the paper that the authors were not able to improve, and these should be added to the limitations paragraph. These include using only HUVEC and not a more physiologically relevant endothelial cell type; seeing little effect at 15 mM glucose; not analyzing NRF2 activation in normal glucose samples; and the limitations of some of the functional measurements, like DAF and the ROS assays. In addition, all these assays were done on endothelial cells *in vitro* in static culture, which may have limited relevance to endothelial cells *in vivo* that are exposed to shear stress from blood flow.

RESPONSE: As any study, our work has both strengths and limitations. Although we believe that addressing in abundant detail every possible limitation may be unusual for this kind of publications, to comply with Reviewer's concerns we have added the following phrases to the manuscript:

- a. To clarify the *in vitro* nature of our experiments, in line 475 we added "cultured" before "endothelial cells", and added "*in vitro*" in Line 38 of the Abstract.
- b. Regarding NRF2 activation in normal glucose samples, in line 491 we have included the statement "...the impact of activating NRF2 in normoglycemia warrants investigation."
- c. In addressing blood flow and the 15 mM glucose concentration, in line 480 we have included: "...**Second, *in vivo*, the vasculature is subjected to multiple stimuli, like shear stress from blood flow, and exposure to glucose fluctuations throughout the day, including exposure to lower pathologic concentrations, such as 15 mM, which in our model didn't induce a robust memory effect.**"
- d. Regarding other endothelial cells, in line 493 we have added "**Lastly, our study focused only on HUVECs; thus, other physiologically relevant endothelial cells or *in vivo* models would have to be investigated**".
- e. With respect to the DAF assays, in line 195 we have included "Although **other methodologies to quantify NO production could be implemented**, these results are indicative..."

We hope these additions effectively address the concerns regarding the limitations of our study.

April 25, 2024

RE: Life Science Alliance Manuscript #LSA-2023-02382-TRR

Dr. Victor Julian Valdes
Universidad Nacional Autónoma de México
Instituto de Fisiología Celular. Cell Biology Department
Cto. Exterior s/n, C.U., Coyoacán
Mexico City 04510
Mexico

Dear Dr. Valdes,

Thank you for submitting your revised manuscript entitled "Reversal of high-glucose-induced transcriptional and epigenetic memories through NRF2 activation". We would be happy to publish your paper in Life Science Alliance pending final revisions necessary to meet our formatting guidelines.

- please be sure that the authorship listing and order is correct
- please upload all figure files as individual ones, including the supplementary figure files; all figure legends should only appear in the main manuscript file
- please add your main, supplementary figure, and table legends to the main manuscript text after the references section
- please add a callout for Figure 5E to your main manuscript text
- the file uploaded as Cover Art appears to be a Graphical Abstract and should be uploaded as such

A. FINAL FILES:

B. MANUSCRIPT ORGANIZATION AND FORMATTING:

Thank you for your attention to these final processing requirements. Please revise and format the manuscript and upload materials within 4 days.

Sincerely,

April 30, 2024

RE: Life Science Alliance Manuscript #LSA-2023-02382-TRRR

Dr. Victor Julian Valdes
Universidad Nacional Autónoma de México
Instituto de Fisiología Celular. Cell Biology Department
Cto. Exterior s/n, C.U., Coyoacán
Mexico City 04510
Mexico

Dear Dr. Valdes,

Thank you for submitting your Research Article entitled "Reversal of high-glucose-induced transcriptional and epigenetic memories through NRF2 activation". It is a pleasure to let you know that your manuscript is now accepted for publication in Life Science Alliance. Congratulations on this interesting work.

DISTRIBUTION OF MATERIALS:

Again, congratulations on a very nice paper. I hope you found the review process to be constructive and are pleased with how the manuscript was handled editorially. We look forward to future exciting submissions from your lab.

Sincerely,
